# The nucleosome DNA entry-exit site is important for transcription termination and prevention of pervasive transcription

A Elizabeth Hildreth[†], Mitchell A Ellison[†], Alex M Francette, Julia M Seraly, Lauren M Lotka, Karen M Arndt*

Department of Biological Sciences, University of Pittsburgh, Pittsburgh, United States

**Abstract** Compared to other stages in the RNA polymerase II transcription cycle, the role of chromatin in transcription termination is poorly understood. We performed a genetic screen in *Saccharomyces cerevisiae* to identify histone mutants that exhibit transcriptional readthrough of terminators. Amino acid substitutions identified by the screen map to the nucleosome DNA entry-exit site. The strongest H3 mutants revealed widespread genomic changes, including increased sense-strand transcription upstream and downstream of genes, increased antisense transcription overlapping gene bodies, and reduced nucleosome occupancy particularly at the 3' ends of genes. Replacement of the native sequence downstream of a gene with a sequence that increases nucleosome occupancy in vivo reduced readthrough transcription and suppressed the effect of a DNA entry-exit site substitution. Our results suggest that nucleosomes can facilitate termination by serving as a barrier to transcription and highlight the importance of the DNA entry-exit site in broadly maintaining the integrity of the transcriptome.

**\*For correspondence:**
arndt@pitt.edu

[†]These authors contributed equally to this work

**Competing interests:** The authors declare that no competing interests exist.

## Introduction

Packaging of the eukaryotic genome into chromatin presents a regulatory barrier to DNA templated processes. Nucleosomes, the fundamental repeating unit of chromatin, are comprised of approximately 147 bp of DNA surrounding an octamer of core histone proteins H2A, H2B, H3, and H4 (*Luger et al., 1997*). To faithfully express protein-coding genes as well as noncoding regions of the genome, RNA polymerase II (Pol II) employs a host of regulatory factors. Among these factors are enzymes that post-translationally modify histones with small chemical moieties (*Lawrence et al., 2016*), histone chaperones that maintain chromatin organization in the wake of Pol II (*Hammond et al., 2017*), and chromatin remodelers that can reposition, exchange, or remove histones from the DNA template (*Clapier et al., 2017*). The mechanisms by which these factors modify chromatin to facilitate or impede transcription initiation and elongation are the subject of much investigation. In contrast, little is understood about how chromatin structure affects transcription termination.

Transcription termination is an essential step in gene expression that is required for the proper 3'-end processing of Pol II transcripts and overall transcriptional fidelity (*Porrua and Libri, 2015*). Unterminated polymerases can continue transcribing into neighboring genes and interfere with their expression (*Greger and Proudfoot, 1998*). Pervasive transcription of eukaryotic genomes (*David et al., 2006*; *van Dijk et al., 2011*; *Xu et al., 2009*) heightens the need for effective transcription termination systems to prevent readthrough transcription into adjacent genes, a problem that is exacerbated in compact genomes such as that of the budding yeast, *S. cerevisiae*. Widespread transcription termination defects can result from cell stress (*Vilborg et al., 2015*) or viral

infection (*Nemeroff et al., 1998*; *Rutkowski et al., 2015*) and are observed in numerous human cancers (*Grosso et al., 2015*; *Kannan et al., 2011*; *Maher et al., 2009*; *Varley et al., 2014*).

In *S. cerevisiae*, there are two major Pol II termination pathways (*Porrua and Libri, 2015*). The cleavage and polyadenylation (CPA) pathway, which functions at protein-coding genes, requires the concerted activities of proteins in the CPF, CF1A and CF1B complexes. Together, these proteins lead to pre-mRNA cleavage at the cleavage and polyadenylation site (CPS) followed by polyA addition to the 3'-end of the cleaved transcript. After pre-mRNA cleavage, Rat1/Xrn2 is thought to promote termination of transcription through capturing the 5' end of the RNA still engaged with Pol II and using its exonuclease activity to catch up to Pol II for eventual disruption of the elongation complex (*Fong et al., 2015*). The Nrd1-Nab3-Sen1 (NNS) termination pathway is primarily dedicated to the termination of short noncoding transcripts in yeast, including small nuclear RNAs (snRNAs), small nucleolar RNAs (snoRNAs), cryptic unstable transcripts (CUTs), Nrd1-unterminated transcripts (NUTs) and aborted transcripts from protein-coding genes (*Porrua and Libri, 2015*; *Arndt and Reines, 2015*; *Schulz et al., 2013*). The specificity of NNS for its targets is governed by the RNA binding activities of Nrd1 and Nab3 (*Carroll et al., 2007*; *Porrua et al., 2012*) and the interaction of Nrd1 with the Pol II C-terminal domain (CTD) phosphorylated at the Ser5 position, a modification enriched in the early transcribed region of genes (*Vasiljeva et al., 2008*). Through kinetic competition with elongating Pol II (*Hazelbaker et al., 2013*), Sen1, a superfamily I helicase, uses its ATPase activity to disrupt the transcription complex (*Porrua and Libri, 2013*). Some NNS-terminated transcripts, such as snoRNAs, are processed to their mature form by the Trf4/Trf5-Air1/Air2-Mtr4 polyadenylation (TRAMP) and nuclear RNA exosome complexes, while others, such as CUTs, are destabilized by these processing factors (*Porrua and Libri, 2015*; *Arndt and Reines, 2015*).

Studies in yeast support a role for chromatin in transcription termination. The *S. cerevisiae* Polymerase-Associated Factor 1 complex (Paf1C) is a conserved, five-subunit protein complex that associates with Pol II on gene bodies and is required for several important transcription-coupled histone modifications, including H2B K123 mono-ubiquitylation, H3 K4 di- and tri-methylation and H3 K36 tri-methylation (*Tomson and Arndt, 2013*; *Van Oss et al., 2017*). Defects in Paf1C and its dependent histone modifications cause terminator readthrough of NNS-dependent snoRNA genes (*Ellison et al., 2019*; *Terzi et al., 2011*; *Tomson et al., 2013*; *Tomson et al., 2011*). In addition, loss of Paf1C subunits leads to changes in polyA site selection at specific protein-coding genes in yeast (*Penheiter et al., 2005*), a phenomenon also observed in mammalian cells (*Yang et al., 2016*). ATP-dependent chromatin remodelers Isw1 and Chd1, which regulate nucleosome spacing within gene bodies and near the CPS, have been linked to transcription termination of some mRNAs (*Alén et al., 2002*; *Morillon et al., 2003*; *Ocampo et al., 2019*). More recent work has uncovered potential physical roadblocks to transcription through genome-wide mapping of transcribing Pol II (*Candelli et al., 2018*) or transcript 3' ends (*Roy et al., 2016*). These studies identified chromatin-associated general regulatory factors (GRFs), such as Reb1, as roadblocks to the progression of Pol II working as either part of the NNS-dependent termination pathway (*Roy et al., 2016*) or as an independent termination process important for blocking readthrough transcription from upstream CPA and NNS terminators (*Candelli et al., 2018*; *Colin et al., 2014*). Although nucleosomes positioned downstream of some NNS-dependent termination sites were proposed to function as roadblock terminators, this hypothesis was not directly tested (*Roy et al., 2016*).

To address the role of nucleosomes in transcription termination, we have performed an unbiased genetic screen to identify histone mutants defective in this process. Our results identify an important role for the nucleosome DNA entry-exit site, a region that controls nucleosome stability and specific histone modifications (*Du and Briggs, 2010*; *Endo et al., 2012*; *Ferreira et al., 2007*; *Li et al., 2005*; *Polach and Widom, 1995*), in preventing transcription terminator readthrough and in broadly controlling pervasive noncoding transcription. Through genome-wide studies and a direct test of the nucleosome roadblock hypothesis, we provide evidence to support the idea that nucleosomes can facilitate transcription termination in vivo.

## Results

### Histone residues at the DNA entry-exit site of the nucleosome are important for transcription termination

To investigate a role for chromatin structure in regulating transcription termination, we performed a genetic screen for amino acid substitutions in histones H3 and H4 that cause defective transcription termination of a reporter construct containing the 70 bp NNS-dependent termination element of the *SNR47* gene (*Carroll et al., 2004*). Briefly, we generated yeast strains lacking the endogenous gene pairs encoding H3 and H4, *HHT1-HHF1* and *HHT2-HHF2*, and containing an integrated copy of the *SNR47* termination reporter or a control reporter lacking the terminator (*Figure 1A*). Using the plasmid shuffle technique (*Sikorski and Boeke, 1991*), we replaced the *URA3*-marked, wild-type H3-H4 expressing plasmid in both strains with *TRP1*-marked plasmids from a comprehensive alanine scanning library of H3 and H4 mutants (*Nakanishi et al., 2008*). Passaged transformants were plated on media lacking histidine to assess the level of transcription occurring at the reporter locus downstream of the *SNR47* terminator (*Figure 1B*).

We identified nine amino acid substitutions in H3 and one amino acid substitution in H4 that cause readthrough of the *SNR47* terminator in the reporter, as measured by growth on media lacking histidine (*Figure 1B*). To confirm the presence of extended transcripts in the histone mutant strains, we performed northern blot analysis of regions downstream of three endogenous snoRNA genes, *SNR47, SNR48,* and *SNR13* (*Figure 1C and D*). The histone mutant strains exhibited varying degrees of terminator readthrough as well as locus-specific effects (*Figure 1D*). Levels of the *SNR48* readthrough product were similar in all the mutants identified in our screen with the exception of the H3 R40A mutant, which failed to show extended products at any of the *SNR* genes tested by northern analysis and is likely a false positive from our screen. In contrast, at the *SNR47* and *SNR13* genes, the H3 T45A and H3 R52A mutants exhibited the highest levels of 3'-extended RNAs. This northern blot analysis validates our reporter-based genetic screen and confirms the transcriptional readthrough defects of eight H3 mutants and one H4 mutant.

Many of the H3 residues altered in the termination-defective mutants localize to the DNA entry-exit site of the nucleosome (*Figure 1E*). The DNA entry-exit site is composed of the H3 αN helix, the preceding H3 tail region, and the H2A C-terminal tail (*Luger et al., 1997*). Since the alanine-scanning H2A library plasmids are marked by *HIS3* (*Nakanishi et al., 2008*), we did not screen for H2A substitutions that cause terminator readthrough using the *SNR47-HIS3* reporter. Instead, we tested the effects of altering specific H2A residues near the DNA entry-exit site on *SNR* gene transcription by northern blot analysis (*Figure 1F*). We observed varying degrees of terminator readthrough in these H2A mutant strains but consistently saw the strongest defects in the H2A H113A, H2A L117A and H2A S121A mutants. The amino acids altered in these H2A mutants map near the positions of the amino acids in H3 that, when mutant, cause strong readthrough phenotypes (*Figure 1D–E*).

### Genome-wide analysis reveals read-through transcription at NNS-terminated loci as a global phenomenon in DNA entry-exit site mutants

To address whether terminator readthrough is a general feature of DNA entry-exit site mutants, we subjected H3 T45A and H3 R52A strains, the strongest mutants as assessed by Northern blot analysis, to strand-specific RNA sequencing analysis (RNA-seq). Pearson's correlation coefficients calculated for each pair of biological replicates for each strain confirm a high degree of reproducibility (*Figure 2—figure supplement 1A–C*). Our spike-in normalized data reveal that transcription at many snoRNA loci is improperly terminated in DNA entry-exit site mutants as measured by a 3' extension index calculation (3'EI) (*Figure 2A–B*). The 3'EI measures fold change in RNA signal 150 bp downstream of the annotated transcription end site (TES) in the mutant compared to wild type (*Tomson et al., 2013*; *Nemec et al., 2017*). By this metric, 30/73 (41.1%) and 43/73 (58.9%) of Pol II-transcribed snoRNAs are read through at least 1.5-fold more than wild type in the H3 T45A mutant and H3 R52A mutant, respectively (*Figure 2A–B*). With respect to their extent of readthrough at all snoRNA genes, the two H3 mutants are strongly correlated (*Figure 2—figure supplement 1D*).

To gain a better understanding of the nature of these readthrough transcripts, we generated de novo annotations from the wild-type and mutant RNA-seq datasets (*Ellison et al., 2019*; *Venkatesh et al., 2016*). De novo annotations were identified as continuous stretches of raw RNA-

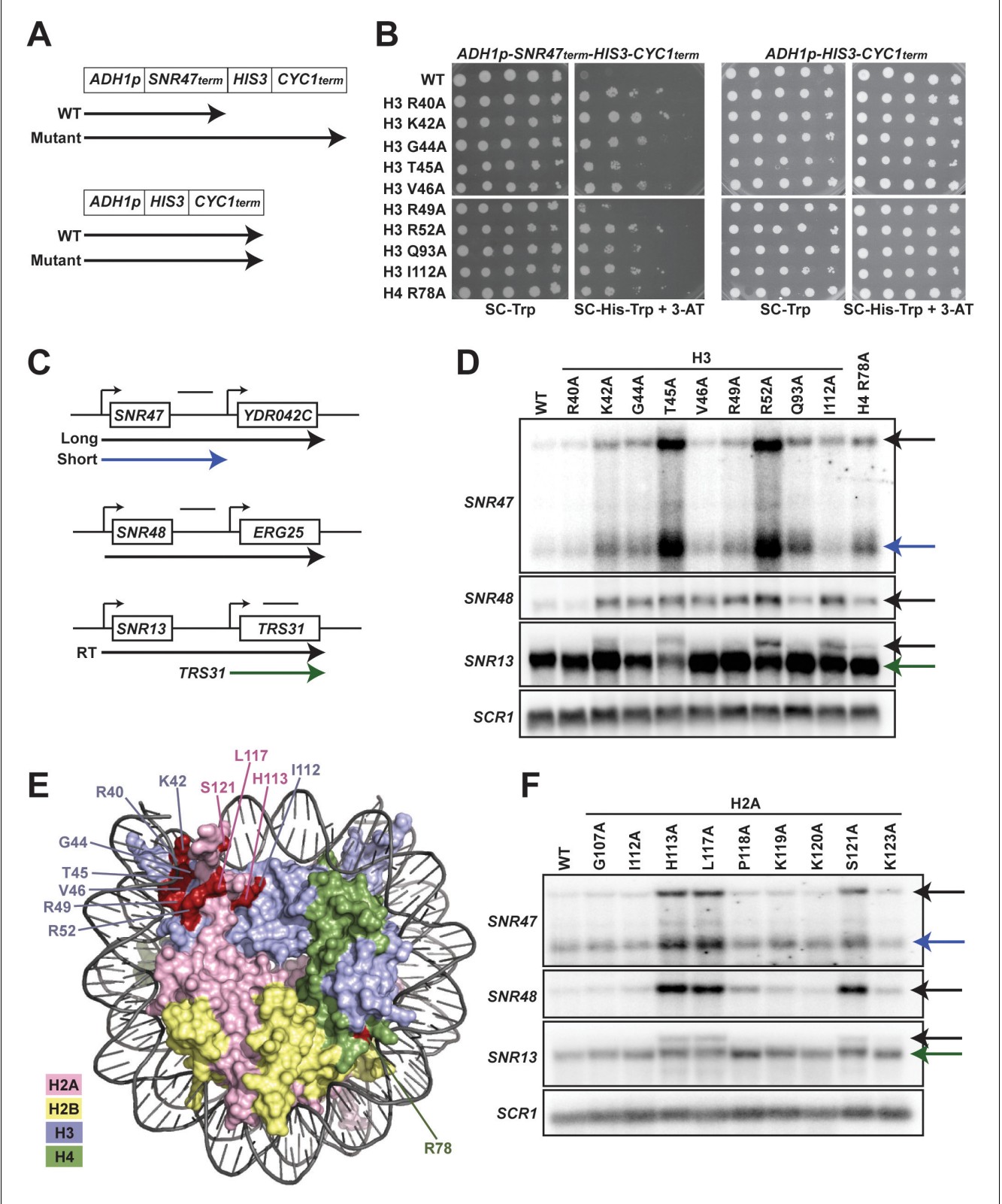

**Figure 1.** Histone residues in the DNA entry-exit site of the nucleosome are important for transcription termination. (**A**) Yeast strains contain either an *SNR47* transcription termination reporter (top, KY3220) or a control transcription cassette lacking the *SNR47* terminator (bottom, KY3219) integrated at the *LEU2* locus. Black arrows denote the transcripts produced from each reporter in wild type (WT) and termination mutant backgrounds. (**B**) Yeast dilution assays to monitor growth of strains expressing the indicated H3 and H4 derivatives as the only source of H3 or H4. Library plasmids (*TRP1-*

*Figure 1 continued on next page*

*Figure 1 continued*

marked, *CEN/ARS*) (*Nakanishi et al., 2008*) expressing the histone gene mutations were introduced by plasmid shuffling into strains expressing the *SNR47* termination reporter (KY3220; left) or the reporter control (KY3219; right). For each strain, a 10-fold dilution series (starting at $1 \times 10^8$ cells/mL) was plated to SC-Trp as a growth control and to SC-His-Trp + 0.5 mM 3-aminotriazole (3-AT), a competitive inhibitor of the *HIS3* gene product, to identify mutants expressing the *HIS3* gene. Plates were incubated at 30°C for 5 days. (C) Diagrams of three snoRNA loci analyzed for termination readthrough by northern analysis. The black bar over each locus denotes the probe position. The intergenic *SNR47* probe detects two read-through transcripts, as indicated by the long black and short blue arrows. The intergenic *SNR48* probe detects a single readthrough transcript. For *SNR13*, the probe overlaps the downstream gene, *TRS31*, and detects a readthrough transcript of *SNR13* (black), as well as the full-length *TRS31* transcript (green). (D, F) Northern blot analysis to assess transcription readthrough of *SNR* genes in (D) H3 and H4 mutants (plasmids shuffled into KY812) and (F) H2A mutants (plasmids shuffled into KY943). Arrows correspond to those shown in the locus diagrams in panel C. *SCR1* serves as the loading control. The northern blots are representative of three independent experiments. (E) X-ray crystal structure of the nucleosome denoting histone residues (highlighted in red) identified in the termination reporter screen and through northern analysis. Due to its buried location, H3 Q93 is not marked. H2A, H2B, H3, and H4 are colored in pink, yellow, lilac, and green, respectively. Structure from PDB ID 1ID3 (*Luger et al., 1997*).

seq reads with sequencing depth of at least 20 bp with gaps no greater than 5 bp. Such an analysis provides empirically supported transcript isoforms based only on the RNA-seq data, not influenced by existing transcript annotations. The de novo annotations reveal 3' extended transcripts at many snoRNA genes, including those confirmed by northern analysis (*Figure 2C–D*).

In addition to snoRNA genes, we asked whether other NNS-dependent ncRNAs are affected in DNA entry-exit site mutants. NUTs are upregulated when the essential termination factor Nrd1 is rapidly depleted from the nucleus (*Schulz et al., 2013*) due to the mechanistic coupling of NNS-dependent termination and ncRNA degradation (*Tudek et al., 2014*). As measured by RNA-seq, levels of NUTs increase by 2.19-fold in the H3 T45A mutant and 2.57-fold in the H3 R52A mutant relative to wild type, arguing that the NNS defect in these histone mutant strains is not specific to *SNR* gene termination (*Figure 2—figure supplement 2A–C*).

## Amino acid substitutions at the nucleosome DNA entry-exit site alter transcription of protein-coding genes

Given the widespread changes observed at Pol II-transcribed ncRNA loci in the H3 T45A and H3 R52A mutants, we extended our analysis to protein-coding genes. We measured fold-change differences in spike-in normalized, stranded RNA-seq read counts over gene bodies, 500 bp upstream of the +1 nucleosome, and 500 bp downstream of the cleavage and polyadenylation site (CPS) (*Figure 3A*). In both mutants, steady-state levels of ORF transcripts only modestly changed compared to the wild type; however, an increase in sense-strand read density both upstream of the +1 nucleosome and downstream of the CPS was apparent. For the region 150 bp downstream of the CPS, the increase in sense-strand RNA levels in the H3 T45A and H3 R52A mutants is modest but statistically significant (*Figure 3B*; *Figure 3—figure supplement 1A*).

To test if changes in RNA levels in the H3 T45A and H3 R52A mutants were due to changes in transcription, we treated cells with 4-thiouracil (4tU) to label nascent RNA and performed sequencing analysis of the 4tU-containing transcripts (*Miller et al., 2011*) (see *Figure 3—figure supplement 2A–C* for replicate comparisons). Interestingly, at 5' and 3' transcript boundaries, an elevation of 4tU-seq signal was observed in both mutants (*Figure 3A*). Compared to the steady-state levels of mRNAs as measured by RNA-seq, we observed a decrease in nascent transcript levels in both histone mutant strains over gene bodies. This effect was especially pronounced in the H3 T45A mutant. For the H3 R52A mutant, we also performed spike-in normalized ChIP-seq analysis of FLAG-tagged Rpb3 (*Figure 3—figure supplement 1B*; *Figure 3—figure supplement 2D-E* for replicate comparisons) and observed a reduction in Pol II occupancy. While other explanations are possible, decreased Pol II occupancy might reflect an increase in Pol II elongation rate as a consequence of disrupted nucleosome structure in the mutants (*Ehrensberger et al., 2013*). The differences in the RNA-seq and 4tU-seq profiles further suggest increased stabilization of transcripts at steady state in the histone mutant strains. This observation is in accordance with evidence showing that global reductions in RNA synthesis are buffered by mechanisms that increase RNA stability (*Timmers and Tora, 2018*). In addition to changes in sense-strand transcription, we also observed that the H3 T45A and H3 R52A substitutions cause increased antisense transcription over genes (*Figure 3—figure supplement 1C*). Increased antisense transcription is consistent with roles of the DNA entry-exit

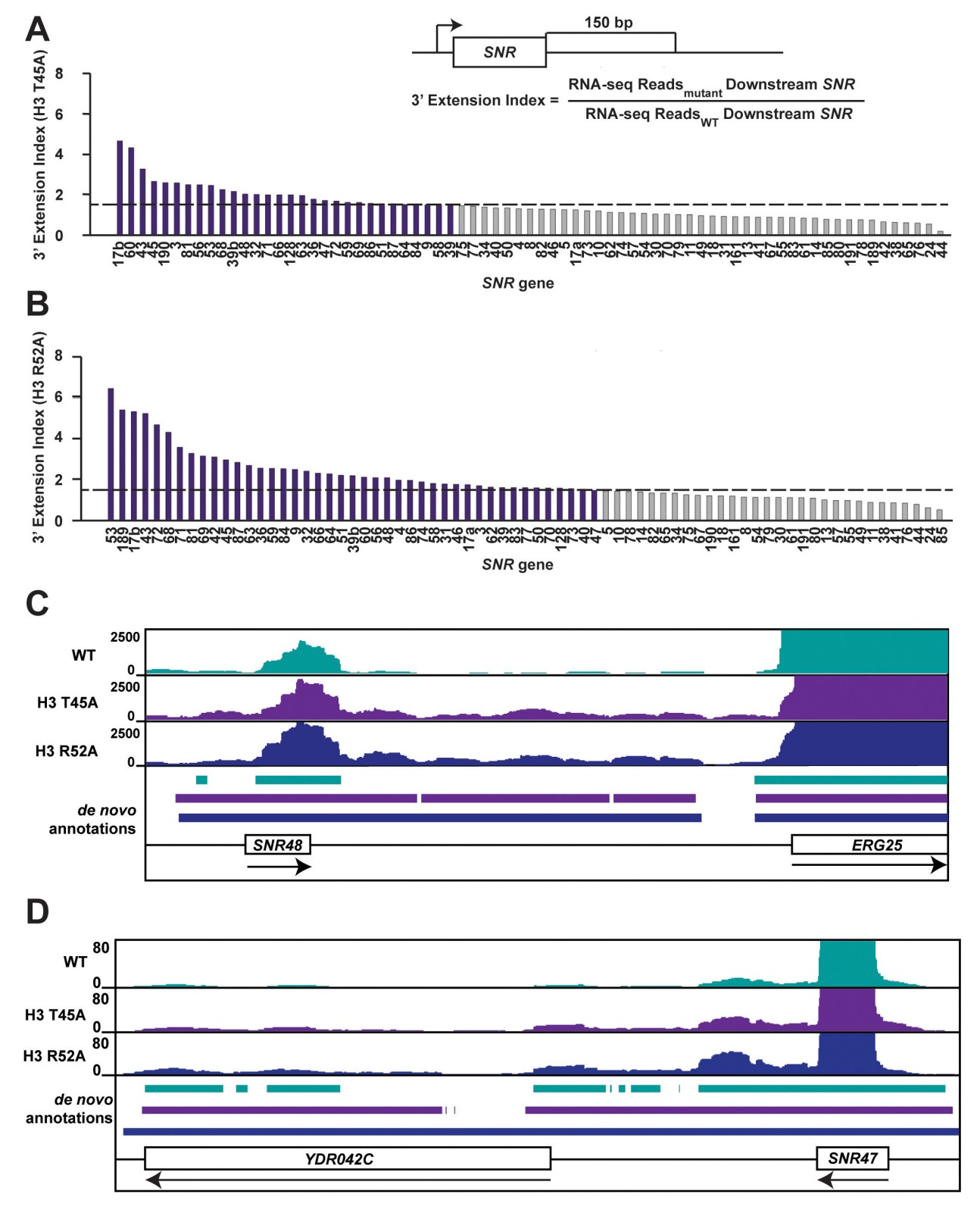

**Figure 2.** Mutations that alter the nucleosome DNA entry-exit site cause widespread 3' extension of snoRNAs. (A, B) 3' extension index in (A) the H3 T45A mutant and (B) the H3 R52A mutant. The ratio (mutant/WT) of spike-in normalized RNA-seq read counts between mutant and wild-type strains produced by plasmid shuffling of strain KY812 was determined in a window 150 bp downstream of each annotated snoRNA 3' end. Ratios equal to or greater than 1.5 (dotted line) are highlighted in purple. (C, D) Browser tracks visualized in IGV (*Thorvaldsdóttir et al., 2013*) showing de novo

*Figure 2 continued on next page*

*Figure 2 continued*

transcript annotations across (**C**) the *SNR48* locus and (**D**) the *SNR47* locus. The browser tracks represent spike-in normalized RNA-seq reads in a wild-type strain and the H3 T45A and H3 R52A mutants. Lines of matching color beneath correspond to the de novo transcript annotations for each dataset. Arrows below gene names indicate directionality of transcription.

The online version of this article includes the following source data and figure supplement(s) for figure 2:

**Source data 1.** *SNR* gene 3′ extension index RNA-seq data.
**Figure supplement 1.** Agreement between biological replicates of RNA-seq datasets.
**Figure supplement 2.** Levels of the Nrd1-unterminated transcripts (NUTs) change in DNA entry-exit site mutants.

site in nucleosome stability and trimethylation of H3 K36 (H3 K36me$^3$) (*Du and Briggs, 2010*; *Endo et al., 2012*; *Ferreira et al., 2007*; *Li et al., 2005*; *Polach and Widom, 1995*), both of which are critical factors in maintaining transcription fidelity and preventing spurious transcription events (*Venkatesh et al., 2016*; *Carrozza et al., 2005*; *Joshi and Struhl, 2005*; *Keogh et al., 2005*; *Venkatesh et al., 2012*; *Kaplan et al., 2003*).

To identify the source of the intergenic nascent transcript density in the H3 T45A and H3 R52A mutants, we mapped the spike-in normalized, stranded 4tU-seq reads in the wild-type and mutant strains to protein-coding genes sorted by their relative orientation (*Figure 4A*). The data support the conclusion that disruption of the DNA entry-exit site leads to elevated transcription both 5′ and 3′ to gene bodies. When viewed as a whole, divergently oriented genes show a greater relative enrichment of nascent transcription 5′ to the coding region and convergently oriented genes show a greater relative enrichment of nascent transcription 3′ to the coding region (*Figure 4B*). At divergently oriented genes, the 5′ nascent transcript enrichment cannot be attributed to terminator readthrough from the neighboring annotated gene. Closer inspection of these genes revealed examples of 5′ extended transcripts (*Figure 4C*, top) as well as examples of readthrough of antisense transcripts originating from within one gene and extending into the intergenic region (*Figure 4—figure supplement 1A*). Enrichment in 3′ read density at convergently oriented genes is most easily explained as an increase in terminator readthrough from at least one gene in the pair. Indeed, for both convergent and tandem gene pairs, we observed clear examples of extended 3′ transcripts that bridge the boundary between an annotated gene and the adjacent intergenic region (*Figure 4C*, middle and bottom). While we observed widespread elevations in intergenic transcripton in the mutants (*Figure 4A–B*), the transcription patterns at some genes appear to be normal (for examples, see *Figure 4—figure supplement 1B*). Additional studies will be needed to elucidate the mechanistic distinctions between genes that are strongly influenced by the DNA entry-exit site substitutions and those that are relatively unaffected by them.

Beyond its role in transcribing mRNAs and structural ncRNAs, such as snoRNAs, Pol II is responsible for the transcription of several classes of ncRNAs that arise from pervasive genome transcription and can play regulatory roles within the cell. Relative to the wild-type strain, our 4tU-seq data revealed increased transcription of most annotated CUTs (*Xu et al., 2009*), stable unannotated transcripts (SUTs; *Xu et al., 2009*), and Xrn1-sensitive unstable transcripts (XUTs; *van Dijk et al., 2011*) in the H3 T45A and H3 R52A mutants (*Figure 4—figure supplement 2*). For some genes, altered transcription of these ncRNAs likely contributes to the increased intergenic signal. However, for others such as the examples shown in *Figure 4C*, the intergenic transcription arises from transcript extensions, and no annotated ncRNAs have been mapped to the intergenic region. Collectively, these results demonstrate that DNA entry-exit site mutants exhibit widespread changes in noncoding transcription in addition to defects in termination and suggest that nucleosomal control of canonical transcription is greatly reduced in these strains.

## Substitution of DNA entry-exit site residues reduces H3 K36me$^3$

Certain residues in the DNA entry-exit site have been implicated in Set2-dependent H3 K36 methylation (*Du and Briggs, 2010*; *Endo et al., 2012*), and mutations in *SET2* impair NNS-dependent termination (*Tomson et al., 2013*). Therefore, we tested whether our DNA entry-exit site mutants exhibit defects in H3 K36me$^3$. In agreement with published results (*Du and Briggs, 2010*; *Endo et al., 2012*), H3 K36me$^3$ levels are greatly reduced in the H3 R49A and H3 R52A mutant strains, while

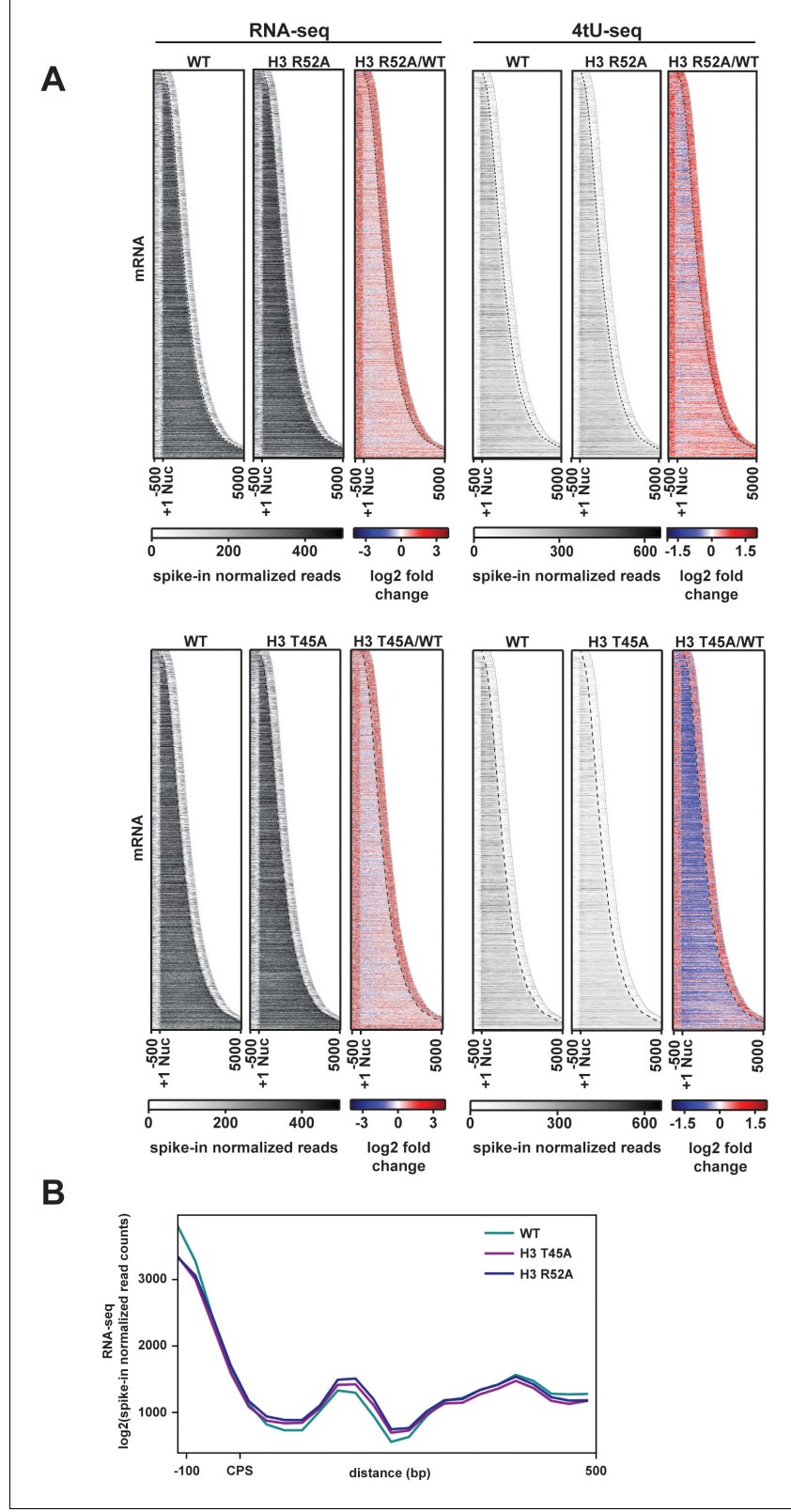

**Figure 3.** Transcriptional changes at protein-coding genes in DNA entry-exit site mutants. (**A**) Heatmaps sorted by gene length and showing spike-in normalized RNA-seq and 4tU-seq read counts (gray scale) and log2-fold change between the H3 R52A or H3 T45A mutant and WT at protein-coding genes. Data are for 6205 protein-coding genes for which positions of +1 nucleosomes, mapped by chemical cleavage (*Brogaard et al., 2012*), and
*Figure 3 continued on next page*

*Figure 3 continued*

cleavage and polyadenylation sites (CPS) (*Ozsolak et al., 2010*) were available. For each gene, 500 bp upstream of the +1 nucleosome to 500 bp downstream of the CPS was plotted in the heatmap. The curved black dotted line marks the position of the CPS. (B) Metagene analysis comparing spike-in normalized read counts in WT and mutant strains over a region from −100 bp to +500 bp from the CPS. Heatmaps and metaplots were generated with deepTools2 (*Ramírez et al., 2014*; *Ramírez et al., 2016*).

The online version of this article includes the following figure supplement(s) for figure 3:

**Figure supplement 1.** Pol II occupancy and antisense transcript levels at protein-coding genes are altered in DNA entry-exit site mutants.

**Figure supplement 2.** Agreement between biological replicates of 4tU-seq and FLAG-Rpb3 ChIP-seq datasets.

other DNA entry-exit site mutants, including the H3 T45A mutant, exhibit a modest decrease in the H3 K36me$^3$ signal (*Figure 5A*).

In addition to its roles in suppressing sense cryptic initiation, H3 K36me$^3$ is also required for repressing transcription of a class of antisense transcripts, termed the Set2-repressed antisense transcripts (SRATs) (*Venkatesh et al., 2016*). We, therefore, analyzed our RNA-seq data for the H3 T45A and R52A mutants for changes in SRAT expression (*Figure 5B*). Both DNA entry-exit site mutants showed upregulation of SRAT expression by greater than four-fold relative to a wild-type control strain (*Figure 5—figure supplement 1*). Despite a more severe H3 K36me$^3$ defect in the H3 R52A mutant, the small increase in SRATs in this mutant over the H3 T45A mutant is not statistically significant.

We noticed that the severity of the termination defects of the H3 T45A and H3 R52A mutants did not correlate with the strength of their H3 K36me$^3$ defects (*Figures 1D* and *5A*). However, given previous connections between Set2 and *SNR* gene termination (*Tomson et al., 2013*), we wanted to test if the transcription defects observed in the DNA entry-exit site mutants were related to their inability to properly establish H3 K36me$^3$. To this end, we used plasmid shuffling to generate an H3 R52A mutant lacking *SET2*. If the defects of the H3 R52A mutant are simply a consequence of losing H3 K36 methylation, then a H3 R52A *set2Δ* double mutant would be expected to have the same phenotype as either single mutant strain. Instead, our results indicate a strong synthetic growth defect in the H3 R52A *set2Δ* double mutant (*Figure 5C*). To rule out the possibility that alternative functions of Set2 led to this genetic interaction and test the effect of removing H3 K36 directly, we constructed a plasmid expressing an H3 K36A, R52A double mutant and introduced it by plasmid shuffling into a yeast strain deleted for the endogenous H3-H4 genes. Unlike the H3 K36A and H3 R52A single mutants, the H3 K36A, R52A double mutant is inviable (*Figure 5D*). We confirmed that the inviability was not due to lack of expression of the double mutant histone by transforming the H3 K36A, R52A-expressing plasmid into yeast cells with tagged endogenous H3 (*HHT1-HA*) and a deletion of the second H3-H4 locus (*hht2-hhf2Δ*). In this context, cells retained the ability to express the double mutant H3 protein (*Figure 5E*). These results argue that the phenotypes of the H3 R52A mutant are, at least in part, independent of its roles in H3 K36me$^3$.

Structural studies implicate histone residues at the DNA entry-exit site in physically interacting with Rtt109, the histone acetyltransferase that catalyzes the modification of H3 at lysine 56 (H3 K56ac) (*Zhang et al., 2018*). This mark on newly synthesized H3 is coupled to histone deposition after DNA repair and replication, replication-independent nucleosome assembly during transcription elongation, and regulation of promoter accessibility during transcription initiation (*Lawrence et al., 2016*). To determine whether residues at the DNA entry-exit site required for transcription termination are also required for H3 K56ac, we assessed the mutant strains for levels of this modification by western blot. Unlike H3 K36me$^3$, our data show that single-residue alanine substitutions in the DNA entry-exit site do not alter levels of H3 K56ac (*Figure 5—figure supplement 2*), suggesting that the transcription defects caused by these substitutions are not due to a general loss of this multifunctional histone modification.

## A mutation that disrupts the DNA entry-exit site causes global changes in nucleosome occupancy

Amino acid substitutions at the DNA entry-exit site are associated with loss of transcription-coupled nucleosome occupancy at specific genes, supporting a role for this nucleosomal surface in restricting

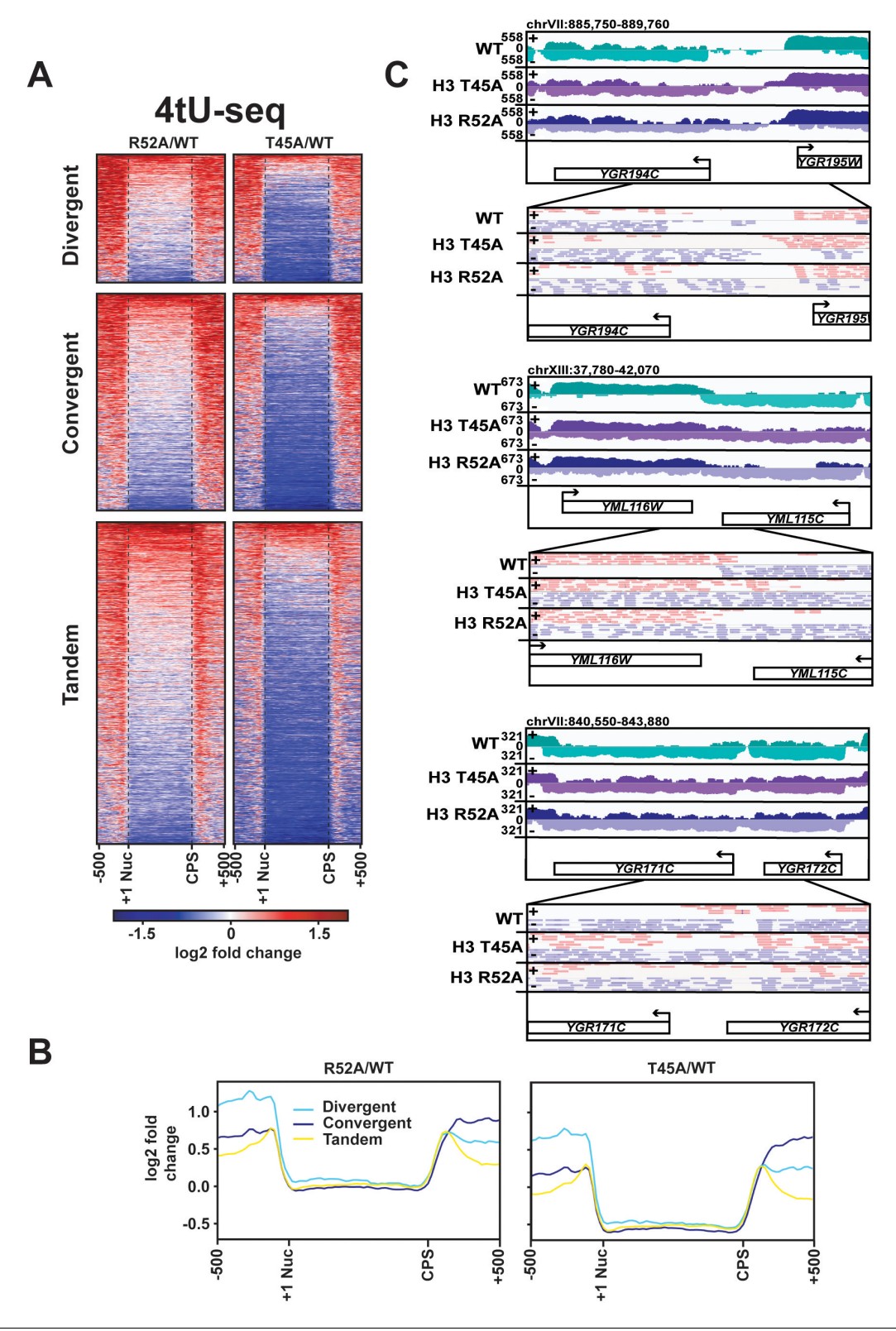

**Figure 4.** DNA entry-exit site mutants display aberrant transcription 5' and 3' to protein-coding genes. (**A**) Heatmaps showing log2-fold change in spike-in normalized 4tU-seq read counts in H3 R52A or H3 T45A mutants relative to WT at divergent, convergent, and tandem genes. Heatmap rows represent 1186 divergent, 1981 convergent, and 2976 tandem protein-coding genes showing 500 bp upstream of the +1 nucleosome (*Brogaard et al., 2012*) and 500 bp downstream of the CPS (*Ozsolak et al., 2010*). (**B**) Metagene plots show averaged intensity over regions displayed in the difference

*Figure 4 continued on next page*

*Figure 4 continued*

heatmaps shown in panel A. (C) Browser tracks visualized in IGV (*Thorvaldsdóttir et al., 2013*) depicting aberrant transcription 5′ and 3′ to divergent (top), convergent (middle), and tandem (bottom) gene pairs in the indicated strains. Data represent log-scaled, spike-in normalized 4tU-seq read density over the plus (+) and minus (-) strands. Regions were chosen to be void of neighboring ncRNA loci. IGV-visualized BAM-file snapshots depict presence or absence of strand-specific read-pairs between genes. Arrows above gene names indicate directionality of transcription. All heatmaps and metagene plots were generated using deepTools2 (*Ramírez et al., 2014*; *Ramírez et al., 2016*) using 25 bp bins.

The online version of this article includes the following figure supplement(s) for figure 4:

**Figure supplement 1.** Some genes in DNA entry-exit site mutants show little to no change in 5′ and 3′ expression.

**Figure supplement 2.** Transcriptional changes at ncRNA loci in DNA entry-exit site mutants.

---

access to DNA (*Hainer and Martens, 2011*; *Hyland et al., 2011*). To test if the H3 R52A substitution globally disrupts nucleosome occupancy, we performed micrococcal nuclease (MNase) treatment of chromatin coupled to deep sequencing (MNase-seq). *S. cerevisiae* chromatin was treated with increasing amounts of MNase to assess bulk nucleosome stability as well as to identify suitable digestion conditions. Based on this assessment, H3 R52A nucleosomes appear more sensitive to MNase than wild-type nucleosomes, as evidenced by reduced levels of di-, tri-, and higher order nucleosomes at comparable treatments (*Figure 6—figure supplement 1A*). MNase-seq analysis was performed on mononucleosomes isolated from the 2.5 U MNase treatments to minimize over-digestion of the mutant and to gain insight on the least chemically perturbed sample available. MNase-seq datasets for the H3 R52A mutant and wild type were spike-in normalized as described in the Materials and methods, compared, and visualized as heatmaps centered on published +1 nucleosome position data (*Brogaard et al., 2012*). Relative to the wild-type control strain, nucleosome occupancy in the H3 R52A mutant is reduced both within coding regions and flanking intergenic regions (*Figure 6A–D*). In the region 150 bp downstream of the CPS, we observed a statistically significant reduction in nucleosome occupancy (*Figure 6B*). A region of reduced nucleosome occupancy over the CPS in the mutant correlates with a region of reduced Pol II occupancy (*Figure 6A,B, D,E*), suggesting that the H3 R52A substitution disrupts nucleosomes which may normally slow Pol II transit in the termination region. In addition to occupancy changes, nucleosome positioning is altered in the mutant with nucleosome peaks shifting downstream of the TSS and widening (*Figure 6C*).

Previous genetic studies in yeast revealed a role for DNA entry-exit site residues in chromosome segregation (*Kawashima et al., 2011*; *Ng et al., 2013*). Indeed, analysis of spike-in normalized read counts across the yeast genome revealed several cases of aneuploidy in the replicate H3 R52A MNase-seq datasets, and presence of these aneuploid chromosomes in the analysis led to only modest agreement between the replicates (*Figure 6—figure supplement 1B–C*). Computational removal of chromosomes that showed evidence of aneuploidy in our MNase-seq datasets greatly improved the correlation (*Figure 6—figure supplement 1D*). To rule out the possibility that changes in chromosome ploidy were affecting our results, we computationally removed all chromosomes with any evidence of aneuploidy in any of our datasets and reanalyzed our MNase-seq, RNA-seq, 4tU-seq and FLAG-Rpb3 ChIP-seq data. Results from the re-analyzed datasets agree with results derived from the original datasets and further support the conclusion that the H3 R52A substitution reduces nucleosome occupancy, alters nucleosome positioning, and leads to increased sense-strand intergenic transcription 5′ and 3′ to gene bodies (*Figure 6—figure supplement 1E*). Anchoring of these data on the CPS revealed a region of reduced nucleosome occupancy upstream and overlapping a region experiencing an apparent increase in transcriptional activity (*Figure 6F*). This increase in RNA signal suggests that the reduction in nucleosome occupancy over the CPS may contribute to transcriptional readthrough of these genes. However, the degree of nucleosome loss in the mutant does not appear to be directly correlated with RNA levels at all genes, consistent with other factors, such as RNA degradation pathways, contributing to the accumulation of 3′-extended transcripts.

In accordance with widespread alterations in nucleosome occupancy and positioning, the H3 R52A mutant and other DNA entry-exit site mutants identified by our genetic screen confer phenotypes indicative of disrupted chromatin structure (*Figure 6—figure supplement 2*). These phenotypes include (i) the <u>S</u>uppressor of <u>T</u>y (Spt⁻) phenotype (*Winston, 1992*), which reports on the ability to bypass the transcriptional effects of transposon insertion mutations in the promoters or 5′ ends of genes; (ii) the <u>B</u>ypass <u>U</u>pstream Activation Sequence (UAS) <u>R</u>equirement (Bur⁻) phenotype

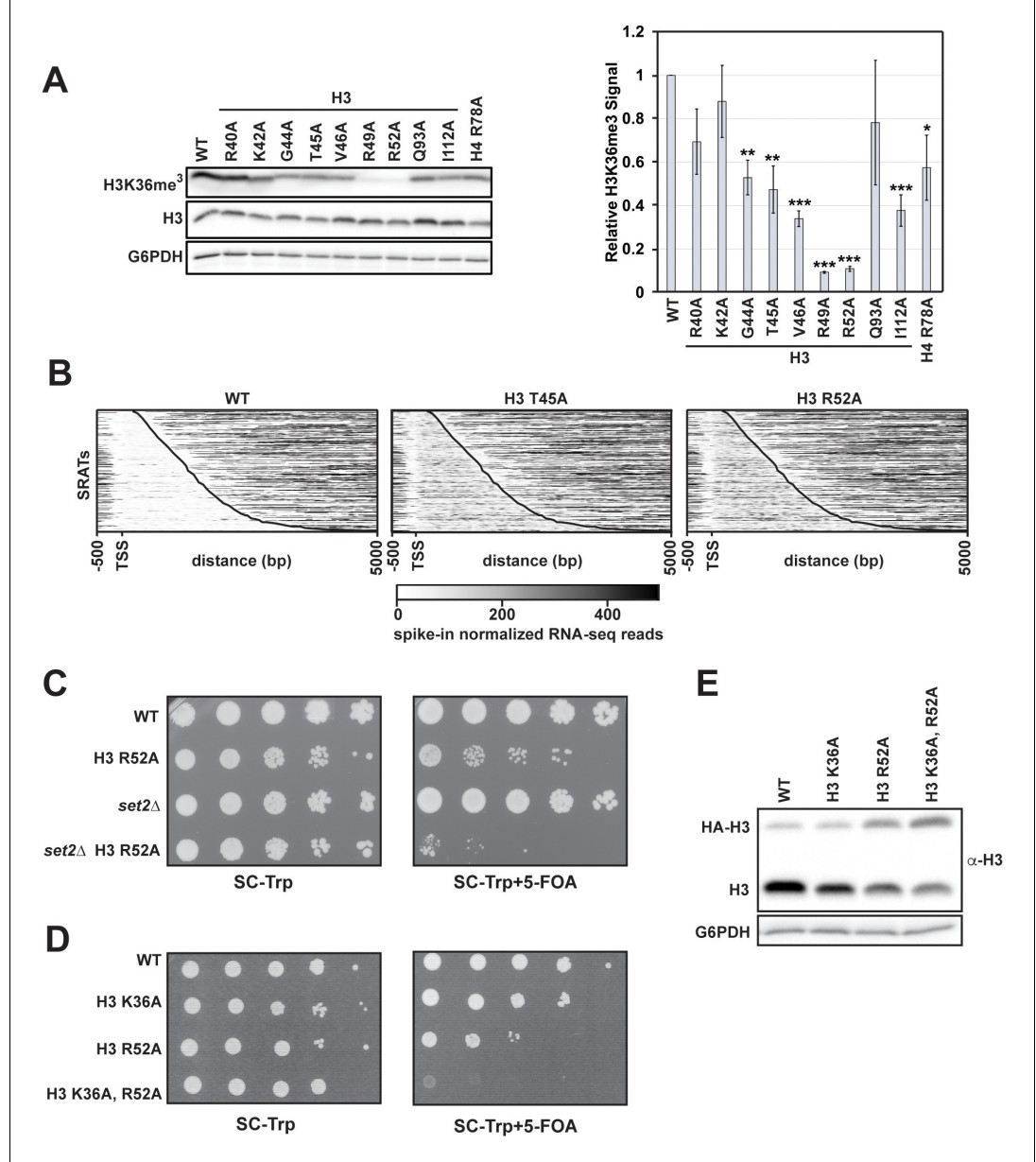

**Figure 5.** H3 K36me³ and the DNA entry-exit site function through genetically distinct pathways. (**A**) Left: western blot analysis of H3 K36me³ levels in H3 and H4 mutant strains. Library plasmids were transformed into KY812 for plasmid shuffling. Extract from a *set2Δ* strain was used to confirm specificity of the H3 K36me³ antibody. Right: Quantification of H3 K36me³ signal after normalizing to total H3 signal. Error bars represent SEM of three biological replicates. Asterisks represent *p<0.05, **p<0.01, and ***p<0.001 by a Student's t-test. (**B**) Heatmaps of spike-in normalized RNA-seq data plotted for regions from −500 bp to +5000 bp relative to the annotated transcription start site (TSS) of Set2-regulated antisense transcripts (SRATS) (**Venkatesh et al., 2016**) in WT and H3 T45A and R52A mutants. Heatmap data are sorted by length of the SRAT annotation and the curved black line marks the annotated TES. (**C**) Plasmid shuffle assay assessing growth of the H3 R52A mutant containing or lacking the *SET2* gene. Cells (10-fold dilution series starting at 1 × 10⁸ cells/mL) were plated to SC-Trp with and without 5-FOA for selection against the *URA3*-marked wild-type H3 plasmid in KY812 (wild-type) and KY3575 (*set2Δ*). (**D**) Plasmid shuffle assay to monitor growth of yeast strains expressing the indicated H3 derivatives as the only source of H3. Yeast strain KY812 was transformed with *TRP1*-marked *CEN/ARS* plasmids expressing the indicated H3 derivatives. Counter selection for the wild-type H3 plasmid was carried out as in C. (**E**) Representative western blot of three biological replicates confirming expression of the plasmid-borne histone mutants analyzed in panel D over integrated HA-tagged, wild-type H3 expressed from the *HHT1* locus (strain KY3511). G6PDH serves as a loading control.

The online version of this article includes the following source data and figure supplement(s) for figure 5:

**Source data 1.** H3 K36me3 western blot data.

**Figure supplement 1.** SRAT expression is significantly increased in DNA entry-exit site mutants compared to wild-type.

*Figure 5 continued on next page*

*Figure 5 continued*

**Figure supplement 2.** Mutations to the DNA entry-exit site do not affect global levels of H3 K56ac.

(*Prelich and Winston, 1993*), which measures expression from the *suc2Δuas(−1900 /- 390)* allele, a mutant gene lacking positive regulatory signals in the promoter; and (iii) the cryptic transcription initiation phenotype, which can be measured using a sensitive reporter that detects aberrant initiation within the *FLO8* coding region, *GAL1pr:FLO8::HIS3* (*Cheung et al., 2008*). However, in contrast to the Spt⁻, Bur⁻ and cryptic initiation phenotypes, the DNA entry-exit site mutants did not exhibit the <u>S</u>WI/SNF <u>In</u>dependent (Sin⁻) phenotype, which measures the ability of histone mutants to overcome the absence of the SWI/SNF chromatin remodeling complex (*Hainer and Martens, 2011*; *Kruger et al., 1995*). This observation suggests that the DNA entry-exit site substitutions affect transcription differently from previously published Sin⁻ mutants, such as H3 V117A (*Hainer and Martens, 2011*). In summary, these growth assays show that DNA entry-exit site mutants exhibit a range of phenotypes associated with altered chromatin structure and transcription.

### Positioning a stable nucleosome downstream of *SNR48* suppresses the termination defect of the H3 R52A mutant

Informed by our MNase-seq data, we hypothesized that nucleosome occupancy at termination sites is important for proper termination by Pol II. To directly test this, we integrated a 133 bp 'superbinder' DNA sequence, which has high affinity for histones (*Wang et al., 2011*), in place of the natural DNA sequence downstream of the *SNR48* termination site, as determined by our de novo transcriptome assembly data (*Figure 7A*). To monitor nucleosome occupancy at this location, we performed H2A ChIP coupled to qPCR using primers conserved in strains containing and lacking the superbinder sequence (*Figure 7B*). These data show that the superbinder sequence increases nucleosome occupancy approximately 3-fold in wild-type cells and 2.5-fold in H3 R52A mutant cells (*Figure 7C*). We note that the difference in nucleosome occupancy observed at the natural sequence between the H3 R52A mutant and the wild-type strain (*Figure 7A*) is not apparent by ChIP qPCR (*Figure 7C*), most likely due to the limited resolution of standard ChIP. Interestingly, the increase in nucleosome occupancy downstream of *SNR48* imparted by the superbinder sequence suppresses the termination defect of the H3 R52A mutant to the level of a wild-type strain lacking the superbinder (*Figure 7D*). The superbinder sequence is also sufficient to suppress basal level transcription readthrough observed in the wild-type strain. We ensured specificity of the readthrough transcript in the context of the superbinder sequence by additionally probing Northern blots for the *SNR48* gene, which can detect the mature *SNR48* transcript and the readthrough transcript (*Figure 7—figure supplement 1A–B*). Together, these data suggest that proper transcription termination is regulated in part by nucleosome occupancy and that the DNA entry-exit site of the nucleosome is important for maintaining proper chromatin structure in termination regions.

## Discussion

Using an unbiased genetic screen of a comprehensive histone mutant library (*Nakanishi et al., 2008*) and a well-established termination reporter (*Carroll et al., 2004*), we identified residues in the DNA entry-exit site of the nucleosome required for transcription termination in vivo. Our detailed analysis of two of the strongest H3 mutants identified in our screen, H3 T45A and H3 R52A, revealed that the appearance of 3'-extended transcripts in these strains is widespread, evident not only at short ncRNAs like snoRNAs but also at some protein-coding genes. Interestingly, while our study was motivated by a desire to probe the role of chromatin in transcription termination, the DNA entry-exit site mutants exhibit broad effects on the transcriptome beyond their effects on termination. Analysis of steady-state and nascent transcript levels revealed increased sense-strand transcription 5' and 3' of protein coding genes, arising from transcript extension or from an increase in overlapping antisense transcription. Moreover, multiple classes of ncRNAs, including CUTs, SUTs, XUTs, NUTs and SRATs, are upregulated in the DNA entry-exit site mutants. Collectively, our findings support a role for chromatin in regulating transcription termination and highlight the importance of the DNA entry-exit site in preventing pervasive transcription genome-wide.

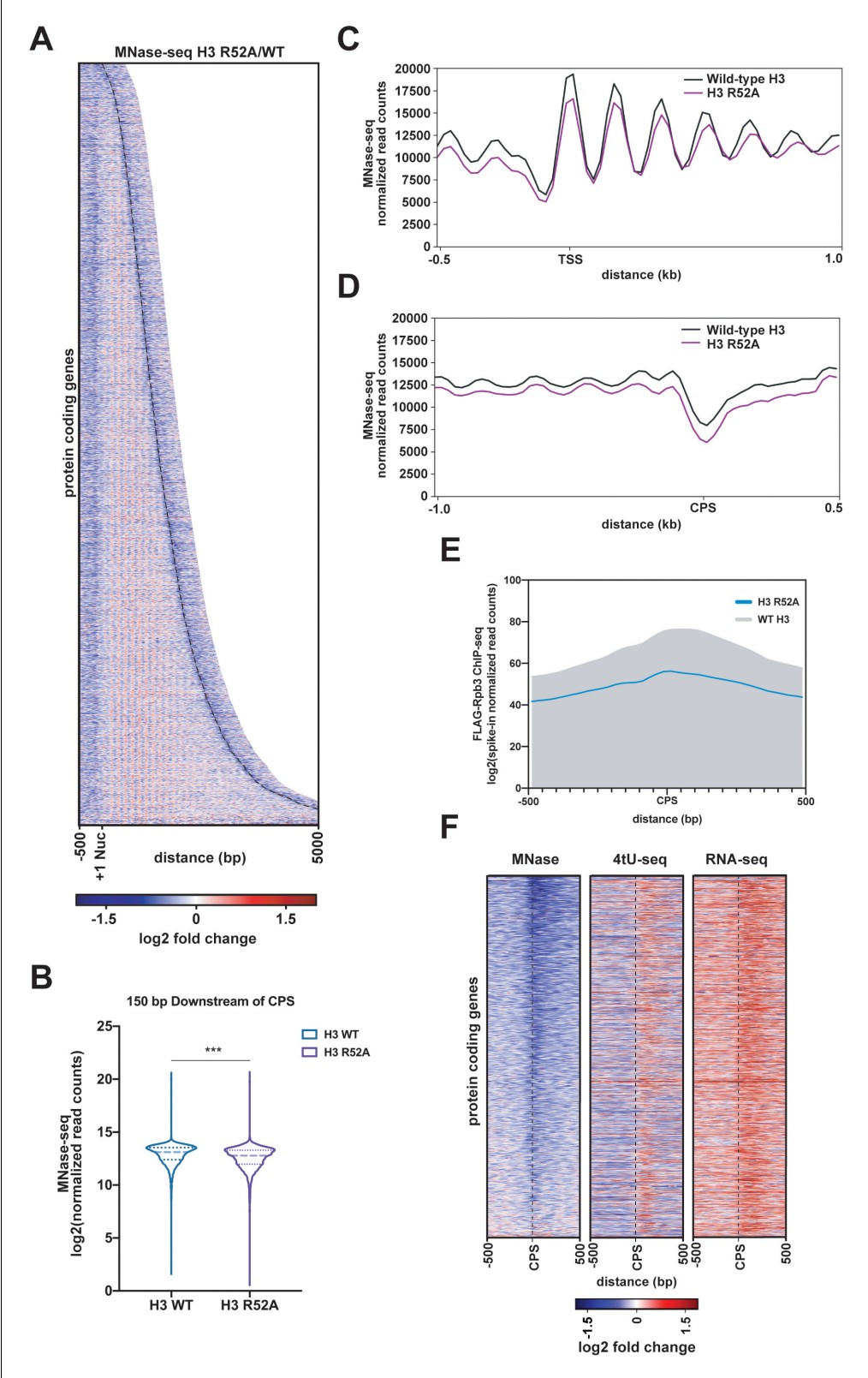

**Figure 6.** Mutation of the DNA entry-exit site alters nucleosomes genome-wide. (**A**) Heatmap of the log2-fold change of MNase-seq read counts (spike-in normalized as described in Materials and methods) of the H3 R52A mutant relative to WT. Rows represent 6205 protein-coding genes (as in *Figure 3*) sorted by length and showing 500 bp upstream of the +1 nucleosome (*Brogaard et al., 2012*) and 500 bp downstream of the CPS (curved black dotted line; *Ozsolak et al., 2010*). (**B**) Violin plot of log2-transformed normalized read counts from MNase-seq data in the region 150 bp

*Figure 6 continued on next page*

*Figure 6 continued*

downstream of the CPS. The difference between WT and the H3 R52A mutant is statistically significant (p<0.0001) as determined by a Wilcoxon rank-sum test. (C) Metagene plots showing normalized MNase-seq read counts for WT (black) and the H3 R52A mutant (purple) in a region from −500 bp to +1000 bp relative to the +1 nucleosome. (D) MNase-seq metagene plots as in C, but plotted from −1000 bp to +500 bp from the CPS. (E) Metagene plots showing spike-in normalized FLAG-Rpb3 ChIP-seq read counts for wild-type (gray) and the H3 R52A mutant (blue) in a region from −500 bp to +500 bp relative to the CPS (*Ozsolak et al., 2010*). All heatmaps and metagene plots were generated using deepTools2 (*Ramírez et al., 2014*; *Ramírez et al., 2016*) using 25 bp bins and 6205 protein-coding genes. All MNase-seq data were produced using a 2.5 U MNase digestion. (F) Heatmaps of MNase-seq, RNA-seq and 4tU-seq data sorted (lowest to highest log2 fold change value based on the MNase-seq data) by mean row value in the MNase-seq data and centered on the CPS and extending up- and downstream by 500 bp. Rows in panel F represent 2879 protein-coding genes present on chrIV, chrVII, chrXII, chrXIV and chrXV (no evidence of aneuploidy in any genomic\transcriptomic dataset for the H3 R52A mutant). The online version of this article includes the following figure supplement(s) for figure 6:

**Figure supplement 1.** Analysis of MNase digestion and MNase-seq data for the the H3 R52A mutant.
**Figure supplement 2.** DNA entry-exit site mutants exhibit chromatin- and transcription-related phenotypes.

Through MNase-seq experiments, we observed a reduction in nucleosome occupancy and a change in nucleosome positioning in the H3 R52A mutant. These observations are consistent with the history of the DNA entry-exit site as a regulator of nucleosome stability (*Li et al., 2005*; *Polach and Widom, 1995*). Amino acid substitutions at the DNA entry-exit site, including H3 R52A and T45A, increase the intrinsic mobility of nucleosomes and DNA unwrapping around the histone octamer in vitro (*Ferreira et al., 2007*). These changes in nucleosome dynamics likely contribute to the changes in transcription and chromatin structure we observe in vivo. Further, nucleosomes impose structural and kinetic barriers to RNA polymerase II progression (*Bondarenko et al., 2006*; *Farnung et al., 2018*; *Kujirai et al., 2018*). We observed reduced Pol II occupancy on coding regions in the H3 R52A mutant, which could be explained by an increase in Pol II elongation rate in the context of disrupted chromatin (*Ehrensberger et al., 2013*). A greater reduction in nucleosome occupancy at the 3' ends of genes in the H3 R52A mutant prompted us to hypothesize that increased transcriptional readthrough might be due to destabilization of a nucleosome roadblock in this region. In support of this idea, integration of a superbinder sequence, which had been previously shown to site-specifically increase nucleosome occupancy in yeast and mammalian cells (*Wang et al., 2011*; *Hainer et al., 2015*), effectively suppressed terminator readthrough at the *SNR48* locus in both wild-type and H3 R52A strains. In designing these experiments, we chose to integrate the superbinder sequence downstream of *SNR48* because, with the exception of H3 R40A, all of the DNA entry-exit site mutants identified in our screen exhibited readthrough transcription at this gene (*Figure 1D*) and because our de novo transcript assembly data mapped the end of the *SNR48* wild-type transcript to a position showing a strong decrease in nucleosome occupancy in the H3 R52A mutant (*Figure 7A*). Previous work on Reb1-mediated roadblock termination noted the presence of Reb1-binding sites downstream of *SNR48* that could aid in NNS-dependent termination (*Roy et al., 2016*) or act as a fail-safe terminator downstream of the NNS terminator (*Candelli et al., 2018*). While our superbinder insertion removed one of three mapped Reb1 binding sites downstream of *SNR48* (*Rhee and Pugh, 2011*), the phenotype of the insertion was to increase, not to decrease, termination, suggesting multiple mechanisms facilitate termination at this gene. These results may not be generalizable to all genes, particularly those in which factors other than nucleosome occupancy or stability dictate termination efficiency. However, they support the idea that nucleosomes can facilitate transcription termination, presumably by slowing Pol II progression and providing time for termination factors to engage and dislodge the elongation complex (*Fong et al., 2015*; *Hazelbaker et al., 2013*).

As an orchestrator of nucleosome stability, the DNA entry-exit site has been recognized as a target for regulatory factors. Proteins that have been proposed to control or capitalize on transient DNA breathing (*Polach and Widom, 1995*) at the DNA entry-exit site include the linker histone H1 (*Frado et al., 1983*; *Riedmann and Fondufe-Mittendorf, 2016*; *Flanagan and Brown, 2016*) and chromatin remodeling factors that may capture nucleosomes in their open state as a primary means of invasion (*Li et al., 2005*; *Ayala et al., 2018*; *Eustermann et al., 2018*; *Li and Widom, 2004*). Previous genetic studies implicated the DNA entry-exit site as a binding site for Set2 (*Du and Briggs, 2010*; *Endo et al., 2012*). More recently, a cryo-EM structure revealed direct binding of Set2 to the αN-helix of H3, including interactions with H3 T45, R49, and R52 (*Bilokapic and Halic, 2019*). H3

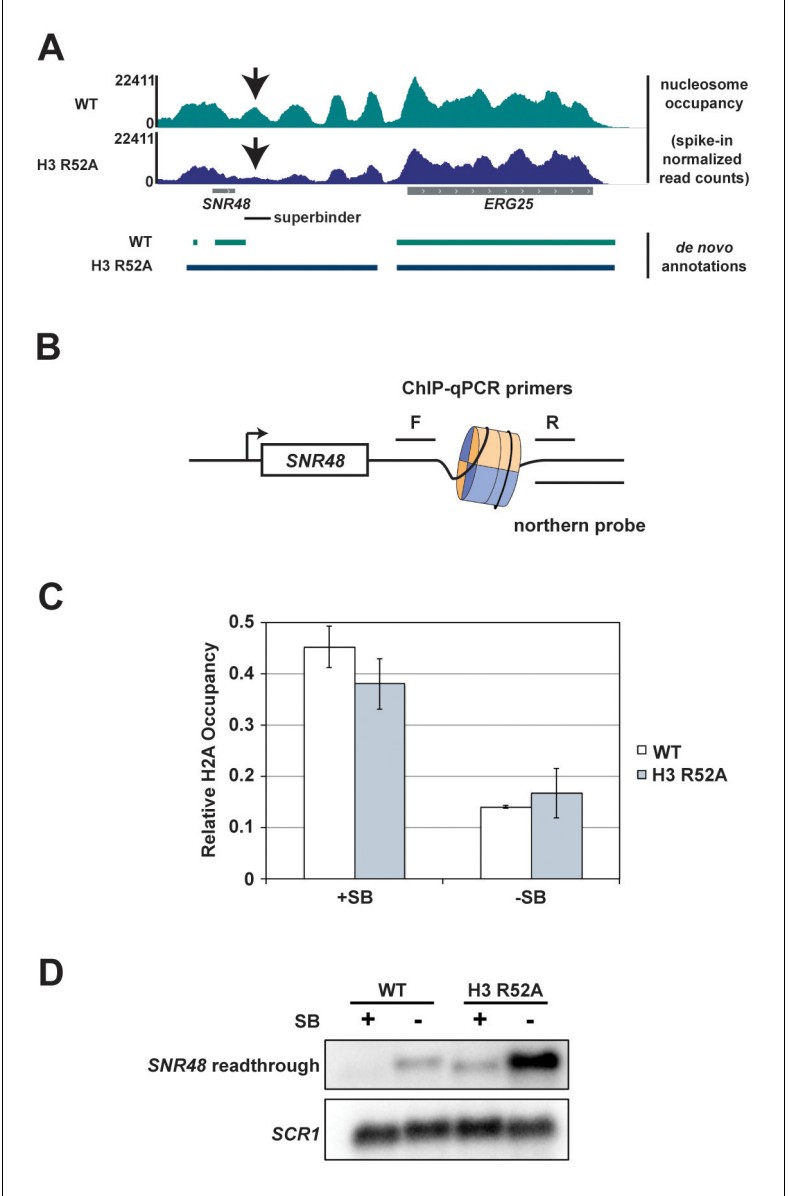

**Figure 7.** Integration of a superbinder DNA sequence downstream of *SNR48* suppresses the termination defect of the H3 R52A mutant. (A) MNase-seq data from a 2.5 U digestion visualized in IGV (*Thorvaldsdóttir et al., 2013*) (top) compared to de novo transcript annotations generated from RNA-seq data from the same strains (bottom). Arrows indicate reduced nucleosome occupancy in the H3 R52A strain compared to the wild-type, which is coincident with the 3' end of the transcript in the wild-type strain. For scale, *SNR48* is 113 bp. The site of integration of the superbinder sequence is indicated by a black bar. (B) Diagram of superbinder nucleosome with locations of the PCR primers for ChIP and the probe used in northern blot analysis to detect *SNR48* readthrough transcription. (C) ChIP-qPCR using an antibody against H2A in KY3221 transformed with either wild-type or mutant histone plasmids. Error bars represent the SEM of three independent biological replicates. (D) Representative northern blot detecting the *SNR48* readthrough transcript in wild-type and H3 R52A strains with or without the superbinder sequence. Northern blot analysis was performed on three independent biological replicates.

The online version of this article includes the following source data and figure supplement(s) for figure 7:

**Source data 1.** H2A ChIP data.

**Figure supplement 1.** Detection of readthrough transcription of *SNR48* by northern analysis in strains with and without the superbinder (SB) sequence.

K36 methylation by Set2 prevents cryptic initiation from intragenic promoters and antisense transcription across gene bodies (*Carrozza et al., 2005*; *Joshi and Struhl, 2005*; *Keogh et al., 2005*; *Venkatesh et al., 2012*; *Kim et al., 2016*). In agreement with earlier genetic studies (*Du and Briggs, 2010*; *Endo et al., 2012*), we found that several of the DNA entry-exit site mutants identified in our termination-based screen are deficient in H3 K36me3 and exhibit phenotypes associated with disrupted chromatin structure. While the reduction in H3 K36me3 likely contributes to the increased antisense transcription in the H3 T45A and H3 R52A mutants (*Venkatesh et al., 2016*) as well as their *SNR* gene termination defects (*Tomson et al., 2013*), we note that these mutants differ significantly in the global levels of this mark and that H3 K36A was not identified as a strong candidate in our termination screen. These observations suggest that the DNA entry-exit site contributes to transcription fidelity through a mechanism distinct from and in addition to promoting H3 K36 methylation. In support of this, we found that the H3 R52A substitution is synthetically lethal with deletion of *SET2* or the H3 K36A substitution, indicating that these H3 residues function, at least in part, through separate pathways.

Several residues at the DNA entry-exit site are post-translationally modified. Acetylation of H3 K56 is involved in nucleosome assembly following DNA replication, repair, and transcription (*Lawrence et al., 2016*). Although structural evidence shows that the acetyltransferase for H3 K56, Rtt109, binds to the DNA entry-exit site near residues identified here (*Zhang et al., 2018*), our mutants do not display global reductions in H3 K56ac, and the H3 K56A mutant was not identified as termination-defective in our screen. Among the residues identified in our screen, H3 K42 can be methylated in *S. cerevisiae* and replacement of this residue with alanine has been reported to cause a 'hypertranscription phenotype' marked by increased intergenic and genic RNA levels (*Hyland et al., 2011*). Our strand-specific RNA-seq and 4tU-seq analyses indicate that the accumulation of intergenic and genic transcripts in the H3 T45A and H3 R52A mutants, and likely the H3 K42A strain (*Hyland et al., 2011*), is due to increased sense-strand transcription 5' and 3' to gene bodies and antisense transcription overlapping gene bodies. H3 T45 has been shown to be phosphorylated in yeast and mammalian cells with regulatory effects on diverse processes including DNA replication, HP1-mediated transcriptional repression and apoptosis (*Baker et al., 2010*; *Hurd et al., 2009*; *Jang et al., 2014*). Interestingly, one study reported elevated H3 T45 phosphorylation downstream of DNA damage response genes by a kinase activated by DNA damage conditions and defective transcription termination at these genes in a mammalian cell line overexpressing H3 T45A (*Lee et al., 2015*). Our data show that the termination defect conferred by the H3 T45A substitution is widespread and is a feature of other DNA entry-exit site mutants.

Together, our findings demonstrate an important role for the nucleosome DNA entry-exit site in maintaining chromatin structure and preventing aberrant transcription genome-wide, including, but not limited to, the readthrough of transcription terminators. Restoration of nucleosome occupancy and transcription termination in a DNA entry-exit site mutant by a nucleosome superbinder sequence suggests that one mechanism by which the DNA entry-exit site regulates termination is through controlling nucleosome stability and imposing a barrier to Pol II progression. However, given the additional importance of the DNA entry-exit site as a target for histone modifiers and chromatin remodelers, it is reasonable to assume that the broad transcriptional effects of DNA entry-exit site substitutions accrue from the combined loss of interactions with multiple regulatory factors as well as potential indirect effects. Moreover, the accumulation of extended and antisense transcripts in these mutants likely imposes a burden on RNA surveillance pathways, raising interesting questions about how these pathways feedback on the transcriptional process. Identifying the mechanisms by which regulatory proteins target the DNA entry-exit site and impact its role in controlling genome access will be an important step in understanding how eukaryotes safeguard against the synthesis and accumulation of aberrant transcripts.

## Materials and methods

### Plasmid construction

Site-directed mutagenesis (QuikChange II kit; Agilent, Santa Clara, CA, #200523) was used to substitute H3 K36 for an alanine residue in the H3 R52A SHIMA library plasmid (*Nakanishi et al., 2008*). A plasmid allowing integration of a histone superbinder (SB) sequence (*Wang et al., 2011*) at any

location within the yeast genome, selectable with a *URA3* marker, was constructed using the pMPY-3xHA backbone (*Schneider et al., 1995*). The SB sequence was amplified by PCR from a TOPO TA vector (*Hainer et al., 2015*) flanked by *NotI* sites or *EcoRI* and *XhoI* sites (primers are listed in *Supplementary file 1*). These fragments and the pMPY-3xHA vector were digested by the appropriate restriction enzymes and ligated together in a stepwise fashion to produce a plasmid containing SB-*URA3*-SB (KB1479), which can be integrated into the yeast genome via standard two-step gene replacement (*Lundblad et al., 2001*). Plasmids were confirmed by DNA sequencing.

## Yeast strains and media

*Saccharomyces cerevisiae* strains used in this study are isogenic to FY2, a *GAL2*[+] derivative of S288C (*Winston et al., 1995*) and are listed in *Supplementary file 2*. Strains were derived by genetic crosses followed by tetrad dissection or by transformation (*Rose, 1987*). *S. cerevisiae* strains were grown in rich medium (YPD) supplemented with 400 µM tryptophan or synthetic complete (SC) media lacking uracil or the specified amino acids. YPSuc and YPGal media contained 1 µg/ml antimycin A and 2% sucrose or 2% galactose, respectively. Unless otherwise noted, transformations were performed with yeast strains lacking endogenous copies of the genes encoding histones H3 and H4 and containing a *URA3*-marked, wild-type H3-H4 plasmid. By selection on SC-Trp medium containing 0.1% 5-fluoroorotic acid (FOA), *URA3* plasmids were shuffled (*Sikorski and Boeke, 1991*) with *TRP1*-marked *CEN/ARS* plasmids expressing alanine-substituted histones H3 and H4 (*hht2-HHF2* or *HHT2-hhf2,* respectively) (*Nakanishi et al., 2008*) or a *TRP1*-marked plasmid expressing the wild-type *HHT2-HHF2* cassette. Complete plasmid shuffling was ensured by sequential passaging of transformants three times on SC-Trp + 5-FOA media. The SB-containing strain was generated by amplifying the SB-*URA3*-SB cassette in two pieces from KB1479. Outside primers (EHO54 and EHO55) had 40 bp of homology to the region downstream of *SNR48,* and universal inside primers (EHO58 and EHO59) annealed to the *URA3* gene. Fragments generated by PCR with EHO54/EHO58 and EHO55/EHO59 overlapped such that upon cotransformation into yeast, homologous recombination joined the two fragments and integrated the full-length cassette, which was confirmed by PCR. After plating to 5-FOA containing media, successful recombination resulted in a single copy of SB located 89 bp downstream of the annotated 3′-end of *SNR48,* replacing bp 609786 through 609883 of chromosome VII. The final recombinant was confirmed by DNA sequencing. *Kluyveromyces lactis* strains were grown in YPD medium supplemented with 400 µM tryptophan and *Schizosaccharomyces pombe* strains were grown in standard YES medium.

## Yeast serial dilution growth assays

Yeast cultures were grown at 30°C and diluted as indicated in the figure legends. Cells were plated by pipetting (3 µl) or using a pinning tool (Sigma-Aldrich, St. Louis, MO, R2383-1EA) on control and selective media to assess specific phenotypes. Plates were incubated at 30°C for at least 3 days and imaged daily.

## Northern blot assays

Total RNA was isolated from log phase yeast cultures ($OD_{600}$ = 0.8–1.0) by hot acid phenol extraction (*Collart and Oliviero, 2001*), and 20 µg of each sample were analyzed by northern blot as described previously (*Swanson et al., 1991*). Double-stranded DNA probes were synthesized by random-prime labeling of PCR fragments with [$\alpha$-$^{32}$P]dATP and [$\alpha$-$^{32}$P]dTTP (PerkinElmer, Waltham, MA). Probe locations are as follows with numbering relative to the start codon of the indicated protein-coding gene, where appropriate: *SNR47-YDR042C* (−325 to −33 of *YDR042C*), *SNR48-ERG25* (−746 to −191 of *ERG25*), *SNR48* (−70 to +92 of the annotated *SNR48* gene), *SNR13-TRS31* (−231 to +449 of *TRS31*), and *SCR1* (−242 to +283 of the annotated *SCR1* gene).

## Western blot assays

Total protein was isolated from log phase yeast cultures ($OD_{600}$ = 0.8–1.0) by bead beating in 20% trichloroacetic acid as described previously (*Cox et al., 1997*). Protein samples were resolved on 15% SDS-PAGE gels and transferred to nitrocellulose (for detection of H3, HA and G6PDH) or PVDF (for detection of H3 K36me$^3$) membrane. Membranes were blocked with 5% milk in TBST (H3, HA and G6PDH) or 3% BSA in PBST (H3 K36me$^3$), and then incubated with primary antibody against

total H3 (*Tomson et al., 2011*, 1:15,000), H3 K36me$^3$ (Abcam, ab9050, 1:1000), HA (Roche, Basel, Switzerland, 12CA5, 1:3000) or G6PDH (Sigma-Aldrich, St. Louis, MO, A9521, 1:20,000), which served as the loading control. For H3 K56ac blots (antisera generous gift of Paul Kaufman, 1:5000), alkaline lysis protein extraction was performed as previously described (*Kushnirov, 2000*). 15% SDS-PAGE gels were run and transferred to nitrocellulose membrane. After incubation with primary antibody, membranes were incubated with a 1:5000 dilution of anti-rabbit (GE Healthcare, Little Chalfont, UK, NA934) or anti-mouse secondary antibody (GE Healthcare, NA931). Proteins were visualized using Pico Plus chemiluminescence substrate (Thermo Fisher, Waltham, MA, #34580) and the ChemiDoc XRS imaging platform (BioRad, Hercules, CA). Signal density for histone post-translational modifications was quantified relative to total H3 signal using ImageJ software, with wild-type signal set to one.

## Chromatin immunoprecipitation and quantitative PCR (ChIP-qPCR)

Chromatin was isolated from 250 mL of yeast cells grown to log phase (OD$_{600}$ = 0.5–0.8) as described previously (*Shirra et al., 2005*). For superbinder (SB) ChIPs sonicated chromatin was incubated overnight at 4°C with an antibody against H2A (ActiveMotif, Carlsbad, CA, 2 μL #39235). Antibody-chromatin complexes were purified with Protein A conjugated to sepharose beads (GE Healthcare, 30 μL GE17-5280-01) for 2 hr at 4°C. After immunoprecipitation, crosslink reversal, and pronase digestion (*Shirra et al., 2005*), DNA was column purified (Qiagen, Hilden, Germany, #28106) and analyzed by qPCR with Maxima 2X SYBR Master Mix (Thermo Fisher, Waltham, MA, K0221). For the SB experiment, H2A occupancy was measured downstream of the *SNR48* locus (+113 to +375 of *SNR48*, across the SB location). All qPCR reactions were performed in biological triplicate and technical duplicate. Protein occupancy was calculated using the appropriate primer efficiency raised to the difference between input C$_t$ and immunoprecipitated C$_t$ values.

## RNA sequencing (RNA-seq)

*S. cerevisiae* and *S. pombe* cells were grown separately to log phase (OD$_{600}$ = 0.8–1.0) and mixed in a 9:1 ratio enabling the use of mapped *S. pombe* reads as an internal spike-in control for RNA-seq analysis. Total RNA was isolated from this mixture of cells by hot acid phenol extraction as described previously (*Collart and Oliviero, 2001*). RNA was DNase treated (Invitrogen, Carlsbad, CA, AM1907) and sent to the Health Sciences Sequencing Core at UPMC Children's Hospital for Ribo-Zero (Illumina) treatment, library preparation, and sequencing.

## 4tU labeling of nascent RNA and sequencing (4tU-seq)

*S. cerevisae* and *S. pombe* nascent RNA was labeled with 4-thiouracil (4tU) as previously described (*Duffy et al., 2015*). Briefly, 4tU was added to log phase cultures (OD600 = 0.8–1.2) to a final concentration of 0.65 mg/mL. Cultures were incubated for 5 min at room temperature with constant agitation. 10 OD units of cells were poured into a half volume of dry-ice cold methanol to rapidly halt metabolic labeling of RNAs (*Barrass et al., 2015*). The resultant mixtures were pelleted for 3 min at 3000 rpm at 4°C, and supernatant was removed prior to snap-freezing the pellets in liquid nitrogen. A RiboPure Yeast RNA extraction kit (Ambion, Austin, TX, AM1926) was used to isolate total RNA from a 9:1 mixture of 4tU-labeled *S. cerevisiae* and *S. pombe* cells. After isolation, 4tU-labeled RNA was biotinylated with 1 mg/mL MTSEA Biotin-XX (Biotium, Fremont, CA, #90066) for 30 min at room temperature in biotinylation buffer (20 mM HEPES, 1 mM EDTA). Meanwhile, streptavidin beads (Invitrogen, #65001) were washed in a high-salt wash buffer (100 mM Tris, 10 mM EDTA, 1 M NaCl, 0.05% Tween-20) and blocked (high-salt wash buffer, 40 ng/uL glycogen) for 1 hr prior to use. A phenol chloroform extraction was used to quench biotinylation reactions by removing unincorporated biotin. Isopropanol precipitated RNA was resuspended in 100 μL nuclease-free water and incubated with blocked streptavidin beads for 15 min. Supernatant was removed and set aside as unlabeled RNA. Beads were washed twice with elution buffer (100 mM DTT, 20 mM HEPES, 1 mM EDTA, 100 mM NaCl, and 0.05% Tween-20) and samples were pooled for concentration via MinElute columns (Qiagen, #74204).

## Microccoccal nuclease sequencing (MNase-seq)

Mononucleosomes were prepared essentially as described (*Wal and Pugh, 2012*). Briefly, cells were grown in SC medium to OD$_{600}$ = 0.8 and crosslinked with formaldehyde at a final concentration of 1%. 100 mL of cells were pelleted, resuspended in FA buffer (50 mM HEPES/KOH, pH 8.0, 150 mM NaCl, 2.0 mM EDTA, 1.0% Triton X-100, and 0.1% sodium deoxycholate), and lysed by bead beating. Cell extracts containing chromatin were pelleted and resuspended in NP-S buffer (0.5 mM spermidine, 0.075% IGEPAL, 50 mM NaCl, 10 mM Tris–Cl, pH 7.5, 5 mM MgCl$_2$, 1 mM CaCl$_2$), and then subjected to digestion by MNase (Thermo Fisher, Waltham, MA, #88216). Mononucleosomal DNA was purified using agarose gel electrophoresis and Freeze N' Squeeze Columns (BioRad, #7326166). As a spike-in control, MNase-treated *K. lactis* DNA, purified in the same manner as *S. cerevisiae* DNA, was added to achieve a 9:1 ratio of *S. cerevisiae* DNA to *K. lactis* DNA. This normalization method reports relative but not absolute differences between samples.

## ChIP and genome-wide sequencing (ChIP-seq)

ChIPs were performed on plasmid shuffled derivatives of KY3232 as described above. *K. lactis* expressing FLAG-tagged Rpb3 from the endogenous locus (*Jin et al., 2017*) was used to prepare chromatin as a spike-in control. Based on DNA content, *S. cerevisiae* chromatin was mixed at a 9:1 ratio with *K. lactis* chromatin. DNA content was determined by treating of 400 µl of chromatin with Pronase at 42°C for 1 hr followed by an overnight incubation at 65°C to reverse the crosslinks. A phenol-chloroform extraction and ethanol precipitation was used to isolate the DNA, which was further purified on a QiaQuick column (Qiagen, 28104) and quantified on a Nanodrop OneC (Thermo Fisher, Waltham, MA). Anti-Flag M2 affinity gel (Sigma-Aldrich, 30 µL A2220) was used to immunoprecipitate FLAG-Rpb3 from the chromatin.

## Library preparation for next-generation sequencing (NGS)

RNA-seq libraries were built by the UPMC Children's Hospital Health Sciences Sequencing Core following rRNA depletion. MNase-seq and ChIP-seq libraries were built prior to sample submission to the sequencing core. Libraries were prepared using the NEBNext Ultra II kit (New England Biolabs, Ipswitch, MA; DNA – E7645) and NEBNext Ultra sequencing indexes (NEB; E7335, E7500, E7710) according to manufacturer's instructions. 4tU-seq libraries were built using a custom SoLo RNA-seq library preparation kit (TECAN, Redwood City, CA; 0516–32) with custom primers targeting *S. cerevisiae* rRNAs for depletion. All RNA and DNA libraries were quantified using Qubit and TapeStation and pooled for paired-end sequencing on an Illumina NextSeq 500 (UPMC Children's Hospital Health Sciences Sequencing Core).

## NGS data processing

Sequencing reads were aligned to the *S. cerevisiae* genome (Ensembl R64-1-1), *S. pombe* genome (Ensembl EF2), or *K. lactis* genome (Ensembl ASM251v1), using HISAT2 (*Kim et al., 2015*) with the following options –no-mixed –no-discordant –no-unal prior to low quality read filtering with the SAMtools suite (*Li et al., 2009*) using the -q option set to 10. 4tU-sequencing reads were aligned with the additional parameters –maxins set to 1000 –min-intronlen set to 52 –max-intronlen set to 1002. The resulting BAM files were used as input to determine read counts for *S. cerevisiae, S. pombe,* and *K. lactis* using either featureCounts (*Liao et al., 2014*) or deepTools2 bamCoverage (*Ramírez et al., 2014*; *Ramírez et al., 2016*) prior to spike-in normalization using the method described in *Orlando et al., 2014*. Generation of bigWig files for browser tracks, and count matrices for graphing and statistical analysis were performed in deepTools2 and R Studio (*RStudio Team, 2016*). For nucleosome mapping, BAM files were used as input for DANPOS2 (*Chen et al., 2013*) using the following options: `–counts –paired` 1 `–pheight` 0.05 `–height` 5 `–testcut` 0.05 `–width` 40 `–distance` 100 `–span` 1 `–smooth_width` 0 `–nor` N `–span` 1 `–mifrsz` 10 `–extend` 73. Wig files generated by DANPOS2 were used to generate BigWig files using the wigToBigWig UCSC utility (*Kent et al., 2010*). For all data types log2 fold change BigWigs were generated using the bigwigCompare command with a bin size of one base pair and a pseudocount of one. Counts were generated for box and whisker plots using the multiBigwigSummary command and plotted using Prism8 or RStudio. Heatmaps and meta-profiles were plotted either directly in deeptools using a combination of the computeMatrix, and either the plotHeatmap or plotProfile command or in Prism8

using data exported from those deeptools commands. For heatmap and meta-profile analyses the computeMatrix command was used to plot data from bigWig files containing spike-in normalized read counts or log2 fold change values over genomic regions specified from a BED file using a bin size of 25 bp and averaging data within each bin.

## De novo transcript analysis

De novo transcript annotations were calculated from strand-specific RNA-seq data using a custom shell script derived from the methods of *Ellison et al., 2019*. To form de novo transcript annotations, this script used a three-step process: 1) RNA-seq read coverage files were created in Bed-Graph format for the plus and minus strands of the yeast genome, 2) regions with depth of coverage greater then 20 were identified, and 3) regions within 5 bp of one another were merged.

## Generation of mRNA BED files and determination of transcription unit orientations

Annotations for mRNA transcriptional units were generated using CDS annotations from SGD and published data for nucleosome positioning and CPS locations (*Brogaard et al., 2012*; *Ozsolak et al., 2010*). Data for +1 nuc, CPS and mRNA annotations were sorted using the sort command in BEDtools (*Quinlan, 2014*; *Quinlan and Hall, 2010*) and CDS annotations were split into those encoded on the plus or minus strand using AWK (*Aho et al., 1979*). The closest command (options: -k 1, -D a, -s, -io, -t first) in BEDtools was then used to select the +1 nuc and CPS closest to the 5' and 3' ends of the CDS annotations from those that existed outside of the boundaries of the original CDS annotation. By this method, a new set of gene annotations was generated which redefined the 5' and 3' boundaries of each locus to those of the closest upstream 5' +1 nuc and downstream 3' CPS, respectively. All scripts used in this analysis as well as input and output files can be found here: https://github.com/mae92/building_annotations_for_mRNAs_in_S.c (*Ellison, 2020*; copy archived at https://github.com/elifesciences-publications/building_annotations_for_mRNAs_in_S.c).

To identify gene pairs in convergent, divergent or tandem orientation, two duplicate files containing the annotations generated using the steps outlined above were compared using the closest command (options: -io, -D a, -k 1) in BEDtools. Using BEDtools closest in this manner allowed for a single closest gene to be identifed for each gene in the yeast genome without allowing overlapping genes to be called. Distances to upstream and downstream genes were reported as negative or positive values (in bp) allowing for files to be further manipulated based on distance and gene orientation. From this list 'close' genes were selected using a distance cutoff of 500 bp. Strand information along with information on positioning (upstream or downstream) mentioned above were used to separate out genes in convergent, divergent and tandem orientations. All scripts used in this analysis as well as input and output files can be found here: https://github.com/mae92/building_annotations_for_mRNAs_in_S.c/tree/master/Orientations.

## Data reproducibility and statistical analysis

Yeast serial dilution growth assays, northern blots, western blots, and ChIPs were performed in biological triplicate with three independently passaged plasmid shuffle transformants. Western blot quantification is shown as the mean signal of three biological replicates with error bars representing the standard error of the mean (SEM). p-Values were calculated via unpaired Student's t-test. ChIP-qPCR was performed in technical duplicate and technical duplicate results were averaged prior to averaging across biological replicates. ChIP-qPCR quantification is shown as the mean value with error bars representing the SEM. NGS experiments – RNA-seq, MNase-seq, 4tU-seq, and Pol II ChIP-seq – were performed in biological duplicate with two independently passaged plasmid-shuffled transformants. Agreement between replicates is shown as biplots with Pearson's correlation. Boxplots of NGS data are presented such that the whiskers indicate the full range of the data and the boxes indicate the second and third quartiles. The dark line within the box indicates the sample median and the distance between the top and bottom of the box represents the interquartile range. For data shown as boxplots p-values were calculated via unpaired Wilcoxon Rank-Sum test.

## Acknowledgements

We thank Fred Winston, Craig Kaplan, and Nathan Clark for yeast strains, Ali Shilatifard for the histone mutant library, Paul Kaufman for anti-H3 K56ac antisera, Jeffry Corden for the *SNR47* termination reporter, and Sarah Hainer for the superbinder-containing plasmid. We are especially grateful to Miler Lee for his assistance with bioinformatic analyses. We thank Christine Cucinotta, James Chuang, and Oliver Rando for technical advice and Fred Winston for critical reading of the manuscript. We are grateful to members of the Arndt lab, especially Margaret Shirra, Brendan McShane and Beth Raupach, and to Miler Lee, Sarah Hainer, and Craig Kaplan, as well as members of their laboratories, for many helpful suggestions. This project used the University of Pittsburgh Health Sciences Sequencing Core at UPMC Children's Hospital of Pittsburgh for all genomic sequencing experiments and was supported in part by the University of Pittsburgh Center for Research Computing through the resources provided. Author publication charges for this article were fully paid by the University Library System, University of Pittsburgh. This work was supported by NIH grant R01 GM052593 to KMA, NIH grant F31 GM129917 to MAE, Andrew Mellon and Margaret A Oweida predoctoral fellowships to AEH, a K Leroy Irvis Fellowship to AMF and a Colella Research Fellowship to L.M.L.

## Additional information

### Funding

| Funder | Grant reference number | Author |
| --- | --- | --- |
| University of Pittsburgh | Andrew Mellon and Margaret A. Oweida graduate fellowships | A Elizabeth Hildreth |
| National Institute of General Medical Sciences | F31 GM129917 | Mitchell A. Ellison |
| University of Pittsburgh | K. Leroy Irvis graduate fellowship | Alex M. Francette |
| University of Pittsburgh | Colella undergraduate research fellowship | Lauren M. Lotka |
| National Institute of General Medical Sciences | R01 GM052593 | Karen M. Arndt |

The funders had no role in study design, data collection and interpretation, or the decision to submit the work for publication.

### Author contributions

A Elizabeth Hildreth, Mitchell A Ellison, Alex M Francette, Conceptualization, Data curation, Software, Formal analysis, Validation, Investigation, Visualization, Methodology, Writing - original draft, Writing - review and editing; Julia M Seraly, Lauren M Lotka, Investigation; Karen M Arndt, Conceptualization, Supervision, Funding acquisition, Writing - original draft, Project administration, Writing - review and editing

### Author ORCIDs

A Elizabeth Hildreth http://orcid.org/0000-0003-1182-6759
Mitchell A Ellison https://orcid.org/0000-0002-5682-096X
Alex M Francette https://orcid.org/0000-0003-1145-5847
Karen M Arndt https://orcid.org/0000-0003-1320-9957

### Decision letter and Author response

Decision letter https://doi.org/10.7554/eLife.57757.sa1
Author response https://doi.org/10.7554/eLife.57757.sa2

# Additional files

## Supplementary files

- Supplementary file 1. Oligonucleotides.
- Supplementary file 2. Yeast strains.
- Transparent reporting form

## Data availability

Sequencing data have been deposited in GEO under accession code GSE147389.

The following dataset was generated:

| Author(s) | Year | Dataset title | Dataset URL | Database and Identifier |
|---|---|---|---|---|
| Hildreth AE, Ellison MA, Francette AM, Seraly JM, Lotka LM, Arndt KM | 2020 | The nucleosome DNA entry-exit site is important for transcription termination and prevention of pervasive transcription | http://www.ncbi.nlm.nih.gov/geo/query/acc.cgi?acc=GSE147389 | NCBI Gene Expression Omnibus, GSE147389 |

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

# Appendix 1

**Appendix 1—key resources table**

| Reagent type (species) or resource | Designation | Source or reference | Identifiers | Additional information |
|---|---|---|---|---|
| Strain, strain background (*Saccharomyces cerevisiae*) | S288C derivatives | This paper; Fred Winston (Harvard Medical School) *Winston et al., 1995* | | See *Supplementary file 2*. Yeast strains |
| Strain, strain background (*Saccharomyces cerevisiae*) | 972 h- | Fred Winston (Harvard Medical School) | FWP10 | spike-in control |
| Strain, strain background (*Saccharomyces cerevisiae*) | NRRL Y-1140 | Nathan Clark (University of Utah) | | spike-in control |
| Genetic reagent, (*Kluyveromyces lactis*) | *KLLA0D16170 g-3XFLAG:: NAT* | *Jin et al., 2017* | YSC193 | spike-in control |
| Recombinant DNA reagent | SHIMA plasmid library | *Nakanishi et al., 2008* | | recombinant histone mutant library |
| Sequence-based reagent | PCR primers | This paper | | See *Supplementary file 1*. Oligonucleo-tides |
| Sequence-based reagent | histone superbinder (SB) sequence | *Wang et al., 2011* | | |
| Recombinant DNA reagent | pMPY-3xHA plasmid | *Schneider et al., 1995* | | Plasmid vector used for SB-*URA3*-SB construct |
| Recombinant DNA reagent | TOPO TA plasmid containing SB sequence | *Hainer et al., 2015* | | Source of SB DNA for construction of SB-*URA3*-SB construct |
| Recombinant DNA reagent | SB-*URA3*-SB plasmid | This study | KB1479 | Used to generate yeast strains with integrated SB |
| Gene (*Saccharomyces cerevisiae*) | *hht2-HHF2* or *HHT2-hhf2* | *Nakanishi et al., 2008* | | |
| Antibody | α-H3 (rabbit polyclonal) | *Tomson et al., 2011* | | western analysis (1:15,000) |
| Antibody | α-H3 K36me$^3$ (rabbit polyclonal) | Abcam | ab9050 | western analysis (1:1000) |
| Antibody | α-HA (mouse monoclonal) | Roche | 12CA5 | western analysis (1:3000) |
| Antibody | α-G6PDH (rabbit) | Sigma-Aldrich | A9521 | western analysis (1:20,000) |
| Antibody | α-H3 K56ac (rabbit) | Paul Kaufman (UMass Medical School) | | western analysis (1:5000) |
| Antibody | α-rabbit IgG-HRP | GE Healthcare | NA934 | western analysis (1:5000) |
| Antibody | α-mouse IgG-HRP | GE Healthcare | NA931 | western analysis (1:5000) |

*Continued on next page*

*Appendix 1—key resources table continued*

| Reagent type (species) or resource | Designation | Source or reference | Identifiers | Additional information |
|---|---|---|---|---|
| Commercial assay or kit | Pico Plus chemilumine-scence substrate | Thermo Fisher | #34580 | |
| Antibody | α-FLAG M2 affinity gel (mouse monoclonal) | Sigma-Aldrich | A2220 | 30 µL α-FLAG beads per 700 µL chromatin (ChIP) |
| Antibody | α-H2A (rabbit polyclonal) | ActiveMotif | #39235 | 1 µL α-H2A per 700 µL chromatin (ChIP) |
| Ather | Protein A conjugated to sepharose beads | GE Healthcare | GE17-5280-01 | 30 µL per chromatin IP |
| Commercial assay or kit | QIAquick PCR purification kit | Qiagen | #28106 | |
| Commercial assay or kit | NEBNext Ultra II library kit for Illumina | New England Biolabs | E7645 | |
| Commercial assay or kit | NEBNext Ultra II sequencing indexes | New England Biolabs | E7335, E7500, E7710 | |
| Commercial assay or kit | SoLo RNA-seq library preparation kit | TECAN | 0516–32 | |
| Commercial assay or kit | Maxima 2X SYBR Master Mix | Thermo Fisher | K0221 | |
| Commercial assay or kit | TURBO DNA-free kit | Invitrogen | AM1907 | |
| Commercial assay or kit | RiboPure Yeast RNA extraction kit | Ambion | AM1926 | |
| Commercial assay or kit | MTSEA Biotin-XX | Biotium | #90066 | |
| Commercial assay or kit | Streptavidin beads | Invitrogen | #65001 | |
| Commercial assay or kit | MinElute columns | Qiagen | #74204 | |
| Commercial assay or kit | QuikChange II kit | Agilent | #200523 | |
| Peptide, recombinant protein | Micrococcal nuclease | Thermo Fisher | #88216 | |
| Sequence-based reagent | [α-$^{32}$P]dATP and [α-$^{32}$P]dTTP | PerkinElmer | | used to generate Northern probes |
| Other | *S. cerevisiae* reference genome | Ensembl | R64-1-1 | |
| Other | *S. pombe* reference genome | Ensembl | EF2 | |
| Other | *K. lactis* reference genome | Ensembl | ASM251v1 | |
| Software, algorithm | HISAT2 | *Kim et al., 2015* | | |
| Software, algorithm | SAMtools | *Li et al., 2009* | | |
| Software, algorithm | RSubread featureCounts | *Liao et al., 2014* | | |

*Continued on next page*

*Appendix 1—key resources table continued*

| Reagent type (species) or resource | Designation | Source or reference | Identifiers | Additional information |
|---|---|---|---|---|
| Software, algorithm | deepTools2 | *Ramírez et al., 2014*; *Ramírez et al., 2016* | | |
| Software, algorithm | R Studio | *RStudio Team, 2016* | | |
| Software, algorithm | DANPOS2 | *Chen et al., 2013* | | |
| Software, algorithm | wigToBigWig UCSC utility | *Kent et al., 2010* | | |
| Software, algorithm | BEDtools | *Quinlan, 2014*; *Quinlan and Hall, 2010* | | |

