## [Decision Letter]

**Acceptance summary:**

All reviewers agree that the authors have done a good job of addressing the previous comments, and that this work provides a significant and careful dissection of how transcription is regulated by histone-DNA interactions at the entry/exit site of a nucleosome.

**Decision letter after peer review:**

Thank you for submitting your article "The nucleosome DNA entry-exit site is important for transcription termination in *Saccharomyces cerevisiae*" for consideration by *eLife*. Your article has been reviewed by three peer reviewers, one of whom is a member of our Board of Reviewing Editors, and the evaluation has been overseen by Cynthia Wolberger as the Senior Editor. The reviewers have opted to remain anonymous.

The reviewers have discussed the reviews with one another and the Reviewing Editor has drafted this decision to help you prepare a revised submission.

As the editors have judged that your manuscript is of interest, but as described below that additional revisions are required before it is published, we would like to draw your attention to changes in our revision policy that we have made in response to COVID-19 (https://elifesciences.org/articles/57162). First, because many researchers have temporarily lost access to the labs, we will give authors as much time as they need to submit revised manuscripts. We are also offering, if you choose, to post the manuscript to bioRxiv (if it is not already there) along with this decision letter and a formal designation that the manuscript is "in revision at *eLife*". Please let us know if you would like to pursue this option. (If your work is more suitable for medRxiv, you will need to post the preprint yourself, as the mechanisms for us to do so are still in development.)

Summary:

While nucleosomes are assumed to act as road blocks for different steps in transcription, their impact on transcription termination has not been systematically tested. The reviewers agreed that this work provides significant and careful experimental evidence towards testing such a role. They found the experiments investigating K36 methylation and K56 acetylation changes to be compelling for ruling out indirect effects. They also felt that the experiments using the nucleosome superbinder DNA sequences helped nail down the role of nucleosome stability. While overall the findings were considered interesting and important for the gene regulation community, the reviewers have some questions about (i) the specificity of the effects on termination vs. cryptic transcription, and (ii) interpretation of the effects in terms of nucleosome occupancy vs. altered nucleosome dynamics. We ask that you address these, and other comments summarized below in a revised version.

Essential revisions:

1) How specifically is this a termination effect vs a general effect of cryptic transcription? The original screen was based on a termination test, but given the increase in the upstream mRNA, anti-sense transcription, and some other ncRNA, we wonder if these mutations simply deregulate transcription in general. If this is the case, the title is a bit misleading. This should be discussed.

2) In terms of mechanism it is not clear how much of the effects are due to lower nucleosome occupancy and how much are due to altered nucleosome dynamics at entry/exit site. It would be ideal to test the effects of some of the key mutants such as H3R52A on DNA breathing and nucleosome stability in vitro and assess if there is a strong correlation between the in vitro and in vivo effects. However, if such experiments are not readily feasible, the authors should explicitly mention in the Discussion the challenge of deconvolving these two effects in vivo.

3) In Figure 2A and B, do these two different mutants generate similar effect on the *SNR* termination? The labeling of the *SNR* names is hard to read. It will be better to generate an overall correlation between the two mutants.

4) Since different genes respond differently to these mutations (like in Figure 2A and B), the authors should divide genes into different groups based on their read-through level and look at the termination signals of these genes (PAS or Nab / Nrd binding sites). If the authors think that the termination effects are related to nucleosome stability, they can also look at the nucleosome positioning pattern near the termination sites in these groups of genes.

5) It was mentioned that the "transcript levels downstream of the CPS were elevated 1.6-fold and 1.9-fold in the H3 T45A and H3 R52A mutants". These numbers seem to be inconsistent with what was shown in Figure 3B.

6) The difference in nucleosome occupancy in Figure 6 seems to be rather minor and very much relies on the spike-in normalization. The statistical significance of such difference should be estimated.

7) The authors proposed a model that these histone mutants affect the elongation rate of the polymerase. Elongation rate change can have many consequences, e.g. change in start site or splicing efficiency. From the RNA-seq data, can the authors find more evidence of increase in the elongation rate?

---

## [Author Response]

Essential revisions:1) How specifically is this a termination effect vs a general effect of cryptic transcription? The original screen was based on a termination test, but given the increase in the upstream mRNA, anti-sense transcription, and some other ncRNA, we wonder if these mutations simply deregulate transcription in general. If this is the case, the title is a bit misleading. This should be discussed.

We agree that the termination defect is only one measure of transcriptional deregulation in the DNA entry-exit site mutants. While the initial premise of the study was to identify histone mutants defective in transcription termination, our comprehensive analysis revealed widespread effects on cryptic and other forms of noncoding transcription. These results highlight the importance of the DNA entry-exit site in controlling the integrity of the transcriptome. We tried to emphasize this better in the revised version. Changes can be found in the wording to the Abstract, Introduction, Results, and Discussion.

In addition, we changed the title of the manuscript to: “The nucleosome DNA entry-exit site is important for transcription termination and prevention of pervasive transcription”

2) In terms of mechanism it is not clear how much of the effects are due to lower nucleosome occupancy and how much are due to altered nucleosome dynamics at entry/exit site. It would be ideal to test the effects of some of the key mutants such as H3R52A on DNA breathing and nucleosome stability in vitro and assess if there is a strong correlation between the in vitro and in vivo effects. However, if such experiments are not readily feasible, the authors should explicitly mention in the Discussion the challenge of deconvolving these two effects in vivo.

The reviewer makes an excellent point. Indeed, the in vivo experiments included in the manuscript cannot distinguish between effects on nucleosome occupancy and effects on nucleosome dynamics. While we agree that a detailed analysis of the dynamics of the histone mutants in vitro is an excellent idea, we feel it is outside the scope of the current study. However, we are fortunate that the lab of Tom Owen-Hughes (MCB 27: 4037) previously studied DNA entry-exit site mutants and showed that nucleosomes harboring the H3 R52A and T45A substitutions exhibit increased sliding rate and increased DNA unwrapping (by a DNA end-to-end FRET assay), relative to wild-type nucleosomes. We have now changed the text in the Discussion to acknowledge the potential role of nucleosome dynamics and also highlight the Owen-Hughes data on both the H3 R52A and H3 T45A substitutions.

3) In Figure 2A and B, do these two different mutants generate similar effect on the SNR termination? The labeling of the SNR names is hard to read. It will be better to generate an overall correlation between the two mutants.

We are grateful for the suggestion. The H3 T45A and H3 R52A mutants agree well in terms of their effects at the *SNR* genes. We calculated the Pearson’s correlation coefficient between 3’ extension indexes for our H3 T45A and H3 R52A mutants which resulted in r = 0.795. We have included a scatter plot showing this result. Please see new Figure 2—figure supplement 1D.

4) Since different genes respond differently to these mutations (like in Figure 2A and B), the authors should divide genes into different groups based on their read-through level and look at the termination signals of these genes (PAS or Nab / Nrd binding sites). If the authors think that the termination effects are related to nucleosome stability, they can also look at the nucleosome positioning pattern near the termination sites in these groups of genes.

In response to this request, we assessed possible relationships between our RNA sequencing results and Nrd1/Nab3 PARCLIP data (Creamer et al. PLOS Genetics, 2011). We also looked for relationships between differential expression in our mutants and snoRNA type and organization using data presented in the Dieci 2009 Genomics review on snoRNAs. We did not observe an obvious relationship between snoRNA readthrough in our mutants and Nrd1/Nab3 binding, snoRNA type or genomic organization. The snoRNA genes were also clustered using our RNA-seq, 4tU-seq, and MNase-seq datasets. While at some *SNR* genes reduced nucleosome occupancy appears to correlate with transcriptional readthrough (an example is shown in Figure 7), this correlation did not extend to all genes. In the section describing the MNase-seq experiments, we included this sentence: “However, the degree of nucleosome loss in the mutant does not appear to be directly correlated with RNA levels at all genes, consistent with other factors, such as RNA degradation pathways, contributing to the accumulation of 3’-extended transcripts.” We also softened the language in the Discussion on this point.

5) It was mentioned that the "transcript levels downstream of the CPS were elevated 1.6-fold and 1.9-fold in the H3 T45A and H3 R52A mutants". These numbers seem to be inconsistent with what was shown in Figure 3B.

We are grateful to the reviewer for pointing this out. After re-analysis, we believe the discrepancy arose from plotting mean transcript levels and reporting median values. We have changed the text in the Results to highlight the modest, but statistically significant effects detected by RNA-seq downstream of the CPS: “For the region 150 bp downstream of the CPS, the increase in sense-strand RNA levels in the H3 T45A and H3 R52A mutants was modest but statistically significant (Figure 3B; Figure 3—figure supplement 1A).”

6) The difference in nucleosome occupancy in Figure 6 seems to be rather minor and very much relies on the spike-in normalization. The statistical significance of such difference should be estimated.

In response to this request, we have generated a violin plot of log2-transformed spike-in normalized read counts from our MNase-seq data in the region 150bp downstream of the CPS and found the difference between WT and the H3 R52A mutant to be statistically significant (p < 0.0001) by a Wilcoxon Rank Sum test. We have included a new figure panel showing this plot (please see new Figure 6B). We have also revised the Materials and methods to clearly state that our spike-in approach for the MNase-seq experiments allows us to detect relative and not absolute changes between samples. Although we did not include these data, we followed up on the MNase-seq experiments with H3 ChIP assays on a small number of loci. We found that the H3 ChIP signals are in agreement with the MNase-seq data in that we do not detect a dramatic loss of nucleosomes on gene bodies. Thus, we do not feel that our MNase-seq analysis normalized out a larger global effect of the H3 R52A mutant.

7) The authors proposed a model that these histone mutants affect the elongation rate of the polymerase. Elongation rate change can have many consequences, e.g. change in start site or splicing efficiency. From the RNA-seq data, can the authors find more evidence of increase in the elongation rate?

In response to this request, we analyzed our 4tU-seq data for potential splicing defects in the H3 R52A and H3 T45A mutants. To do this we calculated intron-exon splice junction ratios (IEJR, fraction of reads that are unspliced) for the WT and mutants. The results revealed a significant splicing defect in the mutants. Specifically, we calculated the median IEJR values for WT, T45A, and R52A to be 0.19, 0.31, and 0.27, respectively (p-values of 4.482e-10 for T45A compared to WT and 5.127e-06 for R52A compared to WT). While these data are supportive of transcription elongation rate defects in the mutants, they could equally be explained as a consequence of a defect in H3 K36 methylation. Tracy Johnson’s lab (Cell Rep. 27:3760) demonstrated a role for Set2, H3 K36 methylation and Eaf3 in regulating splicing in yeast. Since the DNA entry-exit site mutants also have defects in H3 K36 methylation, as shown in Figure 5 of our paper, it is difficult to ascribe the splicing defect to a specific molecular cause. Given this uncertainty, we have chosen not to include our splicing calculations in the revised manuscript but hope to explore the mechanistic basis for these effects in the future.